# Beyond Aggregation: Guiding Clients in Heterogeneous Federated Learning

**Zijian Wang**[1]   **Xiaofei Zhang**[2]   **Xin Zhang**[3]   **Yukun Liu**[4]   **Qiong Zhang**[1*]
[1]Renmin University of China   [2]Zhongnan University of Economics and Law
[3]Meta   [4]East China Normal University

## Abstract

Federated learning (FL) is increasingly adopted in domains like healthcare, where data privacy is paramount. A fundamental challenge in these systems is statistical heterogeneity—the fact that data distributions vary significantly across clients (e.g., different hospitals may treat distinct patient demographics). While current FL algorithms focus on aggregating model updates from these heterogeneous clients, the potential of the central server remains under-explored. This paper is motivated by a healthcare scenario: could a central server not only coordinate model training but also guide a new patient to the hospital best equipped for their specific condition? We generalize this idea to propose a novel paradigm for FL systems where the server actively guides the allocation of new tasks or queries to the most appropriate client. To enable this, we introduce a density ratio model and empirical likelihood-based framework that simultaneously addresses two goals: (1) learning effective local models on each client, and (2) finding the best matching client for a new query. Empirical results demonstrate the framework's effectiveness on benchmark datasets, showing improvements in both model accuracy and the precision of client guidance compared to standard FL approaches. This work opens a new direction for building more intelligent and resource-efficient FL systems that leverage heterogeneity as a feature, not just a bug. Code is available at `https://github.com/zijianwang0510/FedDRM.git`.

## 1 Introduction

Federated learning (FL) has emerged as a powerful paradigm for training machine learning models across distributed data sources without sharing raw data. By enabling clients such as hospitals, financial institutions, or mobile devices to collaboratively train models under the coordination of a central server, FL offers a practical solution for privacy-preserving learning in sensitive domains (Li et al., 2020a; Long et al., 2020; Xu et al., 2021).

A key challenge in applying FL in practice is statistical heterogeneity: clients often hold data drawn from different, non-identically distributed populations. In healthcare, hospitals may serve distinct patient demographics; in finance, banks may encounter different fraud patterns; and on mobile devices, user behavior varies widely. Such heterogeneity can cause local models to drift apart, leading to slower convergence (Li et al., 2020b), biased updates (Karimireddy et al., 2020), and global models that underperform when applied back to individual clients (T Dinh et al., 2020). To address these issues, most FL systems suppress heterogeneity through aggregation corrections, client reweighting, or personalization, leaving the server largely passive as an aggregator of local updates. We argue this misses a key opportunity: the server should actively exploit heterogeneity rather than merely mitigate it.

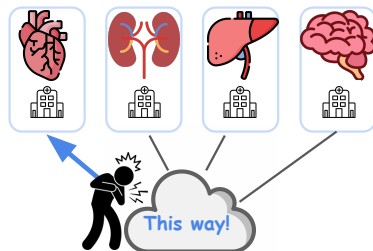

Figure 1: **FL server as an intelligent router**: Leveraging learned data distributions to direct queries to the most specialized client, rather than applying a global model for diagnosis.

---

*Correspondence to: Qiong Zhang (`qiong.zhang@ruc.edu.cn`).

Consider a healthcare scenario: different hospitals may excel at treating different patient groups depending on their location and/or expertise. When a new patient arrives, instead of merely deploying a global model for diagnosis, the server could help identify the hospital best equipped to provide care, leveraging local data distributions to capture specialized expertise. A cartoon illustration of this scenario is given in Fig. 1. Similar opportunities exist in other domains: in finance, the server could direct a fraud detection query to the bank whose historical data best matches the transaction profile; in personalized services, it could route a query to the client with the most relevant user base. These examples illustrate that statistical heterogeneity across clients—often seen as an obstacle—can instead become a valuable resource. They motivate the central insight of our work:

> *Beyond coordinating training, the server can actively exploit client heterogeneity–transforming it from a challenge into a resource by guiding new queries to the most suitable client.*

Much of the existing work in FL has focused on mitigating the challenges of statistical heterogeneity, without using the server for guiding new queries. One major line of research develops aggregation algorithms to reduce the bias induced by non-identically distributed data. Examples include methods that modify local updates before aggregation (Gao et al., 2022; Guo et al., 2023; Zhang et al., 2023), reweight client contributions (Wang et al., 2020; Yin et al., 2024), or introduce regularization terms to align local objectives with the global one (Li et al., 2020b; Acar et al., 2021; Li et al., 2021b). These approaches aim to learn a single global model that performs reasonably well across all clients, but they do not leverage heterogeneity as an asset. A second line of work explores personalization in FL. Rather than enforcing a universal global model, personalization methods adapt models to each client's local distribution (Li et al., 2021d), often through fine-tuning (T Dinh et al., 2020; Collins et al., 2021; Tan et al., 2022; Ma et al., 2022), multi-task learning (Smith et al., 2017; Li et al., 2021c), or meta-learning (Fallah et al., 2020). While these approaches improve local performance, they are typically not designed to address the challenge of guiding new queries or tasks to the most appropriate client. Another related direction is client clustering (Ghosh et al., 2020; Li et al., 2021a; Briggs et al., 2020; Kim et al., 2021; Long et al., 2023), where clients with similar data distributions are grouped and trained jointly within each cluster. This can improve performance under heterogeneity, but still assumes the server's role is limited to coordinating training and distributing models, rather than supporting query routing or task allocation. *Overall, while these approaches are effective for their intended goals, they stop short of enabling the server to actively guide new queries to the most suitable client.*

Motivated by this gap, we introduce a new paradigm in which the FL server not only coordinates training but also learns to guide each incoming query to the client best suited to handle it. Achieving this goal requires two capabilities: (i) effective information sharing across clients despite heterogeneity, and (ii) a principled way to quantify how each client's data distribution differs from the others so that queries can be meaningfully matched. To achieve this, we develop `FedDRM`, a unified framework grounded in density ratio model (DRM) (Anderson, 1979) and empirical likelihood (EL) (Owen, 2001). DRM represents each client's distribution as a *multiplicative density tilt of a baseline distribution*, while EL facilitates nonparametric model learning, enabling the estimation of this baseline distribution in a data-driven manner without parametric assumptions. After profiling out the baseline distribution, the resulting objective decomposes into two interpretable cross-entropy components: one for predicting class labels and another for identifying a sample's client of origin. The first supports standard FL training; the second supplies precisely the signal needed for query-to-client routing, enabling the server to exploit—rather than suppress—statistical heterogeneity.

This formulation leads to three key contributions. *First*, we propose the first statistically grounded FL framework that jointly learns heterogeneous predictive models and the distributional structure required for query routing within a single principled objective. *Second*, we develop a new algorithmic correction for the classification component of the EL objective. Because each client is associated with only a single class label for client identification, the vanilla loss suffers from an extreme form of label shift; we propose a simple yet effective reweighting adjustment that yields a more stable classifier. *Third*, through experiments on benchmark datasets, we demonstrate that our approach consistently improves both predictive accuracy and routing precision compared to standard FL methods, underscoring the benefits of integrating guidance directly into the FL workflow. Together, these developments transform the FL server from a passive aggregator into an intelligent

router capable of directing queries to the most suitable client, opening the door to FL systems that are not only privacy-preserving but also adaptive and expertise-aware.

## 2 FEDDRM: GUIDING CLIENTS IN HETEROGENEOUS FL

### 2.1 PROBABILISTIC DESCRIPTION OF STATISTICAL HETEROGENEITY

Consider an FL system with $m$ clients. Let $\mathcal{D}_i \coloneqq \{(X_{ij}, Y_{ij})\}_{j=1}^{n_i}$ denote the training set on the $i$-th client, where each sample is drawn independently from $P_{X,Y}^{(i)}$. We consider the multi-class classification case where $Y_{ij} \in [K] \coloneqq \{1, \ldots, K\}$ with marginal distribution $\mathbb{P}(Y_{ij} = k) = \pi_{ik}$ for $k \in [K]$, and features conditioned on the labels are distributed as $X_{ij}|(Y_{ij} = k) \sim P_k^{(i)}$. We denote the marginal distribution of the features on client $i$ as $P_X^{(i)}$, and the conditional distribution of $Y$ given $X = x$ as $\{\mathbb{P}^{(i)}(Y = k|X = x)\}_{k=1}^{K}$. Different types of statistical heterogeneity can be described in terms of the family of distributions $\{P_{X,Y}^{(i)}\}_{i=1}^{m}$:

- **Covariate shift**: Clients differ in their marginal feature distributions while sharing the same conditional label distribution. In our notation, this corresponds to

$$P_X^{(i)} \neq P_X^{(i')} \text{ for } i \neq i', \text{ but } \mathbb{P}^{(i)}(Y = k|X = x) = \mathbb{P}^{(i')}(Y = k|X = x) \text{ for all } x \text{ and } k.$$

- **Label shift**: Clients have different label marginals but share the same conditional feature distributions given the label. Equivalently,

$$\pi_i \coloneqq (\pi_{i1}, \ldots, \pi_{iK}) \neq \pi_{i'} \coloneqq (\pi_{i'1}, \ldots, \pi_{i'K}) \text{ for some } i \neq i', \text{ but } P_k^{(i)} = P_k^{(i')} \text{ for all } k.$$

In practice, real-world FL systems often exhibit combinations of these shifts, which leads to the **full distributional shift** where both $\pi_i$ and $\{P_k^{(i)}\}_{k=1}^{K}$ may vary across clients.

### 2.2 A SEMIPARAMETRIC DENSITY RATIO MODEL

For clarity, we begin with the special case of *covariate shift* across clients. Extensions to other types of heterogeneity will then follow naturally. Let $g_\theta(x)$ represent a feature embedding (*e.g.,* an embedding from a DNN parameterized by $\theta$) *s.t.* the conditional distribution of $Y|X$ is given by:

$$\mathbb{P}(Y = k|X = x) = \frac{\exp(\alpha_k + \beta_k^\top g_\theta(x))}{\sum_j \exp(\alpha_j + \beta_j^\top g_\theta(x))}. \tag{1}$$

We drop the superscript $(i)$ since this conditional distribution remains the same across all clients under covariate shift. Applying Bayes' rule to (1), we derive that the class-conditional distributions are connected by an exponential function:

$$dP_k^{(i)}/dP_1^{(i)}(x) = \exp(\alpha_{ik}^\dagger + \beta_k^\top g_\theta(x)) \tag{2}$$

where $dP_k^{(i)}/dP_1^{(i)}$ denotes the Radon–Nikodym derivative of $dP_k^{(i)}$ with respect to $dP_1^{(i)}$ and $\alpha_{ik}^\dagger = \alpha_k + \log(\pi_{i1}/\pi_{ik})$ for $i \in [m]$.

To facilitate knowledge transfer across clients in FL, we assume their datasets share some common underlying statistical structure. Specifically, we relate the client distributions $\{P_1^{(l)}\}_{l=1}^{m}$ through a hypothetical reference measure $P_1^{(0)}$ at the server, using DRM[1] (Anderson, 1979):

$$dP_k^{(i)}/dP_1^{(i)}(x) = \exp(\gamma_i + \xi_i^\top h_\tau(g_\theta(x))) \tag{3}$$

where $h_\tau(\cdot)$ is a parametric function with parameters $\tau$. We refer to $P_1^{(0)}$ as a *hypothetical* reference since the server may not have data directly, although the formulation also applies when server-side

---

[1]DRM provides a framework for modeling the relationship between two or more populations that share similar characteristics. It is highly flexible and encompasses several commonly used parametric distribution families—such as the binomial, exponential, and normal families—as special cases (Kay & Little, 1987).

data are available. The DRM captures differences in the conditional distributions of $X|Y = 1$ across clients via density ratios, with log-ratios modeled linearly in the embeddings. This avoids estimating each distribution separately, focusing instead on relative differences. When the covariate shift is not too severe, the marginal distributions of different clients are connected through this parametric form, making FL effective by leveraging shared structure across clients. On the other hand, if the distributions differ too drastically, combining data from different clients is unlikely to improve performance; in this case, even if the DRM assumption does not hold, it is not a limitation of the formulation but a consequence of the inherent nature of the problem. Thus, the assumption is reasonable in practice. When $\gamma_i = 0$ and $\xi_i = 0$, (3) reduces to the IID case. Under this assumption, we obtain the following relationship between the marginal feature distributions:

**Theorem 2.1.** *With* (2) *and* (3)*, the marginal distributions of $X$ also satisfy the DRM:*

$$dP_X^{(i)}/dP_X^{(0)}(x) = \exp\{\gamma_i^\dagger + \xi_i^\top h_\tau(g_\theta(x))\} \tag{4}$$

*where $\gamma_i^\dagger = \gamma_i + \log(\pi_{i1}/\pi_{01})$ for all $i \in [m]$, and $P_X^{(0)}$ an unspecified reference measure.*

See proof in App. C.1. This theorem relates each client's marginal distribution to the reference distribution through a parametric tilt, which directly facilitates construction of the likelihood. If the reference measure $P_X^{(0)}$ was fully specified, all $\{P_{X,Y}^{(i)}\}_{i=1}^m$ would also be fully determined, and one could estimate the unknown model parameters using a standard maximum likelihood approach. In practice, however, $P_X^{(0)}$ is unknown, and assuming it follows a parametric family risks model mis-specification and potentially biased inference.

To address this challenge, we adopt a flexible, nonparametric approach based on EL (Owen, 2001) that integrate data across heterogeneous populations via DRM (Qin & Zhang, 1997; Fokianos et al., 2001; Chen & Liu, 2013; Li et al., 2017; Liu et al., 2017; 2025). EL constructs likelihood functions directly from the observed data without requiring a parametric form. Instead of specifying a probability model, it assigns probabilities to the observed samples and maximizes the nonparametric likelihood subject to constraints, such as moment conditions. Unlike classical parametric likelihood, EL adapts flexibly to the data, making it particularly suitable when the underlying distribution is unknown or complex, but valid structural or moment conditions are available. Specifically, we let

$$p_{ij} = P_X^{(0)}(\{X_{ij}\}) \geq 0, \quad \forall i \in [m], \ j \in [n_i],$$

treating the $p_{ij}$ as parameters. In this way, the reference measure $P_X^{(0)}$ is represented as an atomic measure without any parametric assumptions, and most importantly *all samples across clients are leveraged for information sharing*. To ensure that $P_X^{(0)}$ and $\{P_X^{(i)}\}_{i=1}^m$ are valid probability measures, the following constraints are imposed:

$$\sum_{i=1}^m \sum_{j=1}^{n_i} p_{ij} = 1, \quad \sum_{i=1}^m \sum_{j=1}^{n_i} p_{ij} \exp\{\gamma_l^\dagger + \xi_l^\top h_\tau(g_\theta(X_{ij}))\} = 1, \quad \forall l \in [m]. \tag{5}$$

### 2.3 A SURPRISINGLY SIMPLE DUAL LOSS

With the semiparametric DRM for heterogeneous FL, we propose a maximum likelihood approach for model learning. Let $p = \{p_{ij}\}$, $\alpha = \{\alpha_k\}$, $\beta = \{\beta_k\}$, $\gamma^\dagger = \{\gamma_i^\dagger\}$, $\xi = \{\xi_i\}$, and $\zeta = (\alpha, \beta, \gamma, \xi, \theta, \tau)$, the log empirical likelihood of the model based on datasets across clients is

$$\ell_N(p, \zeta) = \sum_{i,j} \log P_{X,Y}^{(i)}(\{X_{ij}, Y_{ij}\}) = \sum_{i,j,k} \mathbb{1}(Y_{ij} = k) \log \mathbb{P}(Y = k|X_{ij}) + \sum_{i,j} \log P_X^{(i)}(\{X_{ij}\})$$

$$= \sum_{i,j,k} \mathbb{1}(Y_{ij} = k) \log \mathbb{P}(Y = k|X_{ij}) + \sum_{i,j}\{\gamma_i^\dagger + \xi_i^\top h_\tau(g_\theta(X_{ij})) + \log p_{ij}\},$$

where the last equality makes use of Theorem 2.1. Since our goal is to learn (1) on each client, the weight $p$ becomes a nuisance parameter, which we profile out to learn the parameters that are connected to the conditional distribution of $Y|X = x$. The profile log-EL of $\zeta$ is defined as $p\ell_N(\zeta) = \sup_p \ell_N(p, \zeta)$ where the supremum is under constraints (5). By the method of Lagrange multiplier, we show in App. C.2 that an analytical form of the profile log-EL is

$$p\ell_N(\zeta) = \sum_{i,j,k} \mathbb{1}(Y_{ij} = k) \log \mathbb{P}(Y_{ij} = k|X_{ij}) + \sum_{i,j}\{\gamma_i^\dagger + \xi_i^\top h_\tau(g_\theta(x_{ij})) + \log p_{ij}(\zeta)\} \tag{6}$$

where $p_{ij}(\zeta) = N^{-1}\left\{1 + \sum_{l=1}^m \rho_l \left[\exp\{\gamma_l^\dagger + \xi_l^\top h_\tau(g_\theta(x_{ij}))\} - 1\right]\right\}^{-1}$ and the Lagrange multipliers $\{\rho_l\}_{l=1}^m$ are the solution to

$$\sum_{i,j} \frac{\exp\{\gamma_l^\dagger + \xi_l^\top h_\tau(g_\theta(x_{ij}))\} - 1}{\sum_{l'} \rho_{l'} \left[\exp(\gamma_{l'}^\dagger + \xi_{l'}^\top h_\tau(g_\theta(x_{ij}))) - 1\right]} = 0.$$

Although the profile log-EL in (6) has a closed analytical form, computing it typically requires solving a system of $m$ equations for the Lagrange multipliers, which can be computationally demanding. Interestingly, at the optimal solution these multipliers admit a closed-form expression, yielding a surprisingly simple dual formulation of the profile log-EL presented below.

**Theorem 2.2** (Dual form). *For any fixed $\theta$, the Lagrange multipliers $\rho_l = n_l/N$ and the profile log-EL in (6) becomes*

$$p\ell_N(\zeta) = \sum_{i,j} \log\left\{\frac{\exp(\gamma_i^\ddagger + \xi_i^\top h_\tau(g_\theta(x_{ij})))}{\sum_l \exp(\gamma_l^\ddagger + \xi_l^\top h_\tau(g_\theta(x_{ij})))}\right\} + \sum_{i,j} \log\left\{\frac{\exp(\alpha_{y_{ij}} + \beta_{y_{ij}}^\top g_\theta(x_{ij}))}{\sum_k \exp(\alpha_k + \beta_k^\top g_\theta(x_{ij}))}\right\}$$

*at optimality, up to some constant where $\gamma_i^\ddagger = \log(n_i/n_1) + \gamma_i^\dagger$.*

See App. C.3 for proof. The theorem allows us to define the overall loss function as the negative profile log-EL:

$$\ell(\zeta) = -p\ell_N(\zeta) = \sum_{i,j} \ell_{\text{CE}}(i, h_\tau(g_\theta(x_{ij})); \gamma, \xi) + \sum_{i,j} \ell_{\text{CE}}(y_{ij}, g_\theta(x_{ij}); \alpha, \beta), \quad (7)$$

where $\ell_{\text{CE}}(y, x; \alpha, \beta) = -(\alpha_y + \beta_y^\top x) + \log\{\sum_k \exp(\alpha_k + \beta_k^\top x)\}$ is the cross-entropy loss.

*Remark* 2.3 (**Beyond covariate shift**). Our method is described under covariate shift. The derivations in key steps (2) and (3) do not require the marginal distribution of $Y$ to be identical across clients, which allows us to also accommodate label shift. Importantly, we show that *our approach extends to the more general setting where both $Y|X$ and $X$ differ across clients* in App. D. In this case, after a detailed derivation, we find that the overall loss simplifies to a minor adjustment in the target-class classification head. Concretely, the target-class classification loss is equipped with a client-specific linear head, resulting in the final architecture shown in Fig. 2.

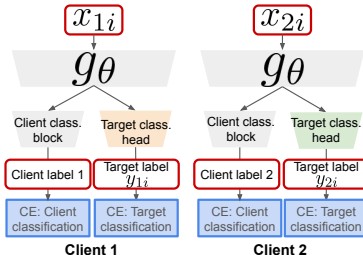

Figure 2: **Network architecture**. Gray blocks are shared among all clients, while colored blocks are specific to each client.

Interestingly, this architecture closely resembles those in personalized FL methods such as (Collins et al., 2021): target-class classification is performed with client-specific heads, while our new client classification component relies on a single shared head across all clients.

*Remark* 2.4 (**Guiding new queries**). Although the derivation is mathematically involved, the resulting loss function is remarkably simple: it consists of two cross-entropy terms, each associated with a distinct classification task. The first term identifies the client from which a sample originates, while the second predicts its target class. The additional client-classification head produces, for each query, a predictive distribution over clients according to (4). By routing a query to the client with the highest predicted probability, we obtain a principled mechanism for assigning new data to the client best equipped to handle it.

## 2.4 OPTIMIZATION ALGORITHM

The overall loss $\ell(\zeta)$ in (7) is defined as if all datasets were pooled together. Since optimizing $\ell(\zeta)$ with vanilla SGD and weight decay is equivalent to minimizing a loss function with an explicit $L_2$ penalty, we denote the loss as $\ell^\rho(\zeta) = \ell(\zeta) + (\rho/2)\|\zeta\|_2^2$, with minimizer $\tilde{\zeta}_N$. The subscript $N$ is used to indicate that this weight is based on $N$ samples. In the FL setting, the global loss decomposes naturally into client-specific contributions: $\ell^\rho(\zeta) = \sum_{i=1}^m (n_i/N)\ell_i(\zeta)$ where

$$\ell_i(\zeta) = \ell_i(\gamma, \xi) + \ell_i(\alpha, \beta) + (\rho/2)\|\zeta\|_2^2,$$

$\ell_i(\gamma, \xi) = n_i^{-1} \sum_j \ell_{\text{CE}}(i, h_\tau(g_\theta(x_{ij})); \gamma, \xi)$, and $\ell_i(\alpha, \beta) = n_i^{-1} \sum_j \ell_{\text{CE}}(y_{ij}, g_\theta(x_{ij}); \alpha, \beta)$.

A key difference arises between these two terms. For the client-classification loss $\ell_i(\gamma, \xi)$, the $i$-th client only observes samples labeled with its own client index $i$. In contrast, the target-class loss $\ell_i(\alpha, \beta)$ typically spans multiple target labels per client (though with varying proportions). This asymmetry leads to more pronounced *gradient drift*[2] in $\nabla \ell_i(\gamma, \xi)$. To illustrate, consider the gradient of the client-classification loss with respect to $\gamma_k$:

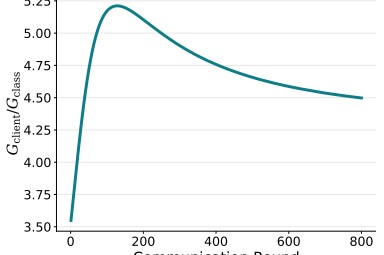

Figure 3: **Relative gradient drift**.

$$\partial \ell_i / \partial \gamma_k = n_i^{-1} \sum_j x_{ij} \{ \mathbb{1}(i = k) - p_k(h_\tau(g_\theta(x_{ij})); \gamma, \xi) \}$$

where $p_k(x; \gamma, \xi) = \exp(\gamma_k + \xi_k^\top x) / \sum_l \exp(\gamma_l + \xi_l^\top x)$. Since $\mathbb{1}(i = k) = 0$ for all $k \neq i$, the gradient contributed by client $i$ provides no meaningful information about other clients' parameters. As a result, local updates to the client-classification head are inherently biased, which in turn amplifies gradient drift relative to target-class head. Fig. 3 shows this effect on a 10-class classification task with 3 clients and a randomly generated embedding using FedAvg: the gradient drift for client classification is markedly more severe than that for target classification.

**Reweighting strategy.** To address this, we draw on reweighting principles (Chen et al., 2018; Liu et al., 2021) to propose a simple yet effective method with theoretical guarantees. Our approach down-weights client classification loss, whose gradient exhibits larger drift, resulting in the per-client loss:

$$\tilde{\ell}_i(\zeta) = (1 - \lambda)\ell_i(\gamma, \xi) + \lambda \ell_i(\alpha, \beta),$$

for $\lambda > 0.5$ and the reweighted global loss is $\tilde{\ell}(\zeta) = \sum_{i=1}^m (n_i/N)\tilde{\ell}_i(\zeta)$, see Algorithm 1.

---

**Algorithm 1:** `FedDRM`

**Input:** Clients $m$, communication rounds $T$, local steps $E$, learning rate $\eta$, trade-off $\lambda$

1   Initialize backbone $\theta^{(0)}$, target head $\{(\alpha_i^{(0)}, \beta_i^{(0)})\}_{i=1}^m$, and client head $(\tau^{(0)}, \gamma^{(0)}, \xi^{(0)})$

2   **for** $t = 0, 1, \ldots, T - 1$ **do**

3      Server broadcasts $(\theta^{(t)}, \tau^{(t)}, \gamma^{(t)}, \xi^{(t)})$ to all clients

4      **for** *client $i \in [m]$ in parallel* **do**

5          $(\theta_i^{(t,0)}, \alpha_i^{(t,0)}, \beta_i^{(t,0)}, \tau_i^{(t,0)}, \gamma_i^{(t,0)}, \xi_i^{(t,0)}) \leftarrow (\theta^{(t)}, \alpha_i^{(t)}, \beta_i^{(t)}, \tau^{(t)}, \gamma^{(t)}, \xi^{(t)})$

6          **for** $k = 0, 1, \ldots, E - 1$ **do**

7              Get target loss $\ell_i(\alpha_i^{(t,k)}, \beta_i^{(t,k)}, \theta_i^{(t,k)})$ and client loss $\ell_i(\tau_i^{(t,k)}, \gamma_i^{(t,k)}, \xi_i^{(t,k)}, \theta_i^{(t,k)})$

8              $\tilde{\ell}_i(\zeta_i^{(t,k)}) \leftarrow \lambda \ell_i(\alpha_i^{(t,k)}, \beta_i^{(t,k)}, \theta_i^{(t,k)}) + (1 - \lambda)\ell_i(\tau_i^{(t,k)}, \gamma_i^{(t,k)}, \xi_i^{(t,k)}, \theta_i^{(t,k)})$

9              $\zeta_i^{(t,k+1)} \leftarrow \zeta_i^{(t,k)} - \eta \nabla \tilde{\ell}(\zeta_i^{(t,k)})$

10          **end**

11          $(\theta_i^{(t+1)}, \alpha_i^{(t+1)}, \beta_i^{(t+1)}, \tau_i^{(t+1)}, \gamma_i^{(t+1)}, \xi_i^{(t+1)}) \leftarrow (\theta_i^{(t,E)}, \alpha_i^{(t,E)}, \beta_i^{(t,E)}, \tau_i^{(t,E)}, \gamma_i^{(t,E)}, \xi_i^{(t,E)})$

12          Client $i$ sends $(\theta_i^{(t+1)}, \tau_i^{(t+1)}, \gamma_i^{(t+1)}, \xi_i^{(t+1)})$ back to the server

13      **end**

14      Server updates $(\theta^{(t+1)}, \tau^{(t+1)}, \gamma^{(t+1)}, \xi^{(t+1)}) \leftarrow \sum_{i=1}^m \frac{n_i}{N}(\theta_i^{(t+1)}, \tau_i^{(t+1)}, \gamma_i^{(t+1)}, \xi_i^{(t+1)})$

15   **end**

---

To accelerate convergence, a larger value of $\lambda$ is desirable. However, as $\lambda \to 1$, the target-class classification begins to dominate, which hinders effective training of the client classification and ultimately weakens the model's ability to guide clients. To illustrate the trade-off between accuracy and convergence, we consider a simplified setting where the embedding is fixed (*i.e.*, $\theta$ and $\tau$ are known) and the true data-generating mechanism follows a multinomial logistic model with parameters $\zeta^{\text{true}} = (\gamma^{\text{true}}, \xi^{\text{true}}, \alpha^{\text{true}}, \beta^{\text{true}})$. We define the heterogeneity measure $G^2(\zeta) = \sum_{i=1}^m (n_i/N)\|\nabla \ell_i(\zeta) - \nabla \ell(\zeta)\|_2^2$, which admits the decomposition $G^2(\zeta) = (1 - \lambda)^2 G_{\text{client}}^2(\gamma, \xi) + \lambda^2 G_{\text{class}}^2(\alpha, \beta)$. Let $\bar{G}^2$, $\bar{G}_{\text{client}}^2$, and $\bar{G}_{\text{class}}^2$ denote the corresponding maximum values across updating rounds $t = 0, 1, \ldots, T - 1$. Then, $\bar{G}^2 \leq (1 - \lambda)^2 \bar{G}_{\text{client}}^2 + \lambda^2 \bar{G}_{\text{class}}^2$. With this notation in place, we state the following result:

---

[2]The gradient drift of the client loss is $G_{\text{client}}^2 := \sum_{i=1}^m (n_i/N)\|\nabla \ell_i(\gamma, \xi) - \sum_{l=1}^m (n_i/N)\nabla \ell_l(\gamma, \xi)\|^2$, and that of the target-class loss is $G_{\text{class}}^2 := \sum_{i=1}^m (n_i/N)\|\nabla \ell_i(\alpha, \beta) - \sum_{l=1}^m (n_i/N)\nabla \ell_l(\alpha, \beta)\|^2$.

**Theorem 2.5.** *Assume $\ell^\rho$ is $\mu$-strongly convex and $L$-smooth. Suppose $\eta \leq 1/L$ and furthermore $\eta L E \leq 1/4$. Let $\zeta^{(t)}$ be the output after $t$ communication rounds. Then as $T, N \to \infty$ we have*

$$\|\zeta^{(T)} - \zeta^{true}\|_2^2 = O_p\left(\frac{\{(1-\lambda)\|\mathcal{I}_\gamma\|_{\min} + \rho\}^{-1} + \{\lambda\|\mathcal{I}_\beta\|_{\min} + \rho\}^{-1}}{N} + \frac{\eta^2 E^2 \{(1-\lambda)^2 \bar{G}_{client}^2 + \lambda^2 \bar{G}_{class}^2\}}{1 - (1-\eta\mu)^E}\right)$$

*where $\|A\|_{\min} = \lambda_{\min}(A)$, and $\mathcal{I}_{client}$ and $\mathcal{I}_{class}$ denote the Fisher information matrices with respect to $(\gamma, \xi)$ and $(\alpha, \beta)$, respectively.*

The proof and the detailed definition of Fisher information matrix is deferred to App. E. The first term in the bound capture the statistical accuracy, while the last term reflects the convergence rate. For faster convergence, a larger $\lambda$ is preferred, while for higher accuracy, $\lambda$ must be chosen to balance $\{(1-\lambda)\|\mathcal{I}_\gamma\|_{\min} + \rho\}^{-1}$ and $\{\lambda\|\mathcal{I}_\beta\|_{\min} + \rho\}^{-1}$. Together, these terms reveal the trade-off role of $\lambda$. In practice, since the Fisher information matrices and gradient drifts are unknown, $\lambda$ can be tuned using a validation set. We empirically demonstrate the trade-off in Fig. 4.

## 3 EXPERIMENTS ON BENCHMARK DATASETS

### 3.1 EXPERIMENT SETTINGS

**Datasets.** We conduct experiments on CIFAR-10 and CIFAR-100 (Krizhevsky, 2009), each containing $60,000$ $32 \times 32$ RGB images. CIFAR-10 has 10 classes with $6,000$ images per class. CIFAR-100 has 100 classes, with 600 images per class, grouped into 20 superclasses. Based on these datasets, we construct three tasks of increasing complexity: (a) 10-class classification on CIFAR-10, (b) 20-class classification using the CIFAR-100 superclasses, and (c) 100-class classification using the fine-grained CIFAR-100 labels.

**Non-IID settings.** Since standard benchmark datasets do not inherently exhibit statistical heterogeneity, we simulate non-IID scenarios following common practice (Wu et al., 2023; Tan et al., 2023; Lu et al., 2024). We introduce both label and covariate shifts. For **label shift**, we construct client datasets using two partitioning strategies: (1) *Dirichlet partition with $\alpha = 0.3$ (Dir-0.3)*: Following (Yurochkin et al., 2019), we draw class proportions for each client from a Dirichlet distribution with concentration parameter $\alpha = 0.3$, leading to heterogeneous label marginals and unequal dataset sizes across clients. (2) *$S$ shards per client ($S$-SPC)*: Following (McMahan et al., 2017), we sort the data by class, split it into equal-sized, label-homogeneous shards, and assign $S$ shards uniformly at random to each client. This yields equal dataset sizes while restricting each client's label support to at most $S$ classes. Each dataset is first partitioned across clients using one of the partitioning strategies, and within each client, the local dataset is further split $70/30$ into training and test sets. For **covariate shift**, all three nonlinear transformations are applied to each client's dataset: (1) *gamma correction*: brightness adjustment with client-specific gamma factor $\gamma$. (2) *hue adjustment*: color rotation with client-specific hue factor $\Delta h$. (3) *saturation scaling*: color vividness adjustment with client-specific saturation factor $\kappa$. We set $\gamma \in \{0.6, 1.4\}$, $\Delta h \in \{-0.1, 0.1\}$, and $\kappa \in \{0.5, 1.5\}$ in the main experiment, resulting in an 8-client setting. See examples in App. F.1.

**Baselines.** We compare our `FedDRM` against a variety of state-of-the-art personalized FL techniques, which learn a local model on each client. Ditto (Li et al., 2021c) encourages local models to stay close via global regularization. FedRep (Collins et al., 2021) learns a global backbone with local linear heads. FedBABU (Oh et al., 2022) freezes local classifiers while training a global backbone, then fine-tunes classifiers per client. FedPAC (Xu et al., 2023) personalizes through feature alignment to a global backbone. FedALA (Zhang et al., 2023) learns client-wise mixing weights that adaptively interpolate between the local and global models. FedAS (Yang et al., 2024) aligns local weights to the global model, followed by client-specific updates. ConFREE (Zheng et al., 2025) resolves conflicts among client updates before server aggregation. We also compare with other standard FL algorithms– FedAvg (McMahan et al., 2017), FedProx (Li et al., 2020b), and FedSAM (Qu et al., 2022)–which aim to achieve a single global model under data heterogeneity. To ensure fair comparison, we fine-tune their global models locally on each client, yielding personalized variants denoted FedAvgFT, FedProxFT, and FedSAMFT.

**Network architecture.** We use ResNet-18 (He et al., 2016) as the feature extractor (backbone), which encodes each input image into a 512-dimensional embedding. For the baselines, this embedding is projected to 256 dimensions via a linear layer and fed into the image classifier. `FedDRM`

extends this design by adding a separate client-classification head: the 512-dimensional embedding is projected to 256 dimensions and fed into the client classifier. Importantly, FedDRM uses the same image classification architecture as all baselines.

**Training details.** To ensure fair comparison, all methods are trained for 800 communication rounds with 10 local steps per round and a batch size of 128. For fine-tuning-based methods, we allocate 700 rounds for global training and 100 rounds for local fine-tuning. We use SGD with momentum 0.9, an initial learning rate of 0.01 with cosine annealing, and weight decay $5 \times 10^{-4}$. Method-specific hyperparameters are tuned to achieve their best performance.

## 3.2 EVALUATION PROTOCOL

To assess the effectiveness of our proposed method in guiding clients under heterogeneous FL, we introduce a new performance metric, termed **system accuracy**. This metric is designed to evaluate the server's ability to guide clients effectively. Concretely, we construct a pooled test set from all clients. For FedDRM, we first use the client classification head to identify the most likely client for each test sample by maximizing the client classification probability. The local model of the selected client is then used to predict the image class label. For baseline methods, which lack this client-guidance mechanism, we instead apply a majority-voting strategy: each client's personalized model makes a prediction for every sample in the pooled test set, and the majority label is taken as the final prediction. The overall classification accuracy on the pooled test set is reported as the system accuracy. We also report the widely used **average accuracy** in personalized FL, which measures each local model's classification accuracy on its own test set. The final value is computed as the weighted average across all clients, with weights proportional to the size of each client's training set. In all experiments, we report the mean and standard deviation of both average accuracy and system accuracy over the final 50 communication rounds.

## 3.3 MAIN RESULTS

We present the system accuracy and average accuracy in Tab. 1 and Tab. 2, respectively. Across

Table 1: System accuracy on CIFAR-10/20/100 under Dir-0.3 and 5/25-SPC settings.

| Method | CIFAR-10 | | CIFAR-20 | | CIFAR-100 | |
|---|---|---|---|---|---|---|
| | Dir-0.3 | 5-SPC | Dir-0.3 | 25-SPC | Dir-0.3 | 25-SPC |
| Ditto | $47.64 \pm 0.25$ | $46.99 \pm 0.23$ | $29.56 \pm 0.18$ | $31.87 \pm 0.16$ | $15.97 \pm 0.15$ | $19.51 \pm 0.16$ |
| FedRep | $24.96 \pm 0.19$ | $33.19 \pm 0.22$ | $23.83 \pm 0.15$ | $24.82 \pm 0.20$ | $11.11 \pm 0.12$ | $12.02 \pm 0.12$ |
| FedBABU | $57.43 \pm 0.17$ | $57.17 \pm 0.24$ | $36.96 \pm 0.17$ | $40.78 \pm 0.13$ | $22.92 \pm 0.17$ | $27.27 \pm 0.15$ |
| FedPAC | $25.14 \pm 0.21$ | $33.24 \pm 0.19$ | $23.83 \pm 0.17$ | $24.83 \pm 0.18$ | $11.17 \pm 0.12$ | $11.99 \pm 0.12$ |
| FedALA | $61.33 \pm 0.17$ | $53.20 \pm 0.20$ | $32.78 \pm 0.17$ | $35.79 \pm 0.14$ | $20.70 \pm 0.14$ | $25.80 \pm 0.16$ |
| FedAS | $28.76 \pm 0.19$ | $39.71 \pm 0.20$ | $27.16 \pm 0.16$ | $27.50 \pm 0.15$ | $13.87 \pm 0.13$ | $13.51 \pm 0.14$ |
| ConFREE | $25.66 \pm 0.22$ | $34.06 \pm 0.22$ | $24.08 \pm 0.17$ | $25.15 \pm 0.18$ | $11.32 \pm 0.13$ | $12.12 \pm 0.13$ |
| FedAvgFT | $54.90 \pm 0.22$ | $56.19 \pm 0.17$ | $37.53 \pm 0.18$ | $41.17 \pm 0.16$ | $25.21 \pm 0.16$ | $27.96 \pm 0.17$ |
| FedProxFT | $55.01 \pm 0.20$ | $56.27 \pm 0.21$ | $37.61 \pm 0.18$ | $41.20 \pm 0.15$ | $25.15 \pm 0.13$ | $27.82 \pm 0.18$ |
| FedSAMFT | $55.83 \pm 0.21$ | $51.73 \pm 0.19$ | $34.23 \pm 0.14$ | $36.60 \pm 0.17$ | $22.97 \pm 0.16$ | $26.89 \pm 0.15$ |
| FedDRM | $\mathbf{62.78 \pm 0.20}$ | $\mathbf{58.50 \pm 0.23}$ | $\mathbf{37.63 \pm 0.21}$ | $\mathbf{41.44 \pm 0.19}$ | $\mathbf{26.03 \pm 0.15}$ | $\mathbf{31.24 \pm 0.17}$ |

all settings, FedDRM consistently outperforms the baselines on both metrics, demonstrating its ability to leverage statistical heterogeneity for system-level intelligence while also providing effective client-level personalization. In contrast, the baselines primarily focus on addressing data heterogeneity, resulting in lower system accuracy due to disagreements among their personalized models. Additionally, when using a majority-vote approach as an intelligence router, baseline methods must evaluate all $m$ local models, whereas FedDRM requires evaluating only a single model. The shared backbone in FedDRM can also be efficiently repurposed for image prediction by feeding it into the corresponding client-specific classification head. We also compare the influence of label shift in this experiment beyond covariate shift, the results align with our expectation that the less severe label shift Dir-0.3 case has a higher accuracy than 5-SPC for all methods.

Table 2: Average accuracy on CIFAR-10/20/100 under Dir-0.3 and 5/25-SPC settings.

| Method | CIFAR-10 | | CIFAR-20 | | CIFAR-100 | |
|---|---|---|---|---|---|---|
| | Dir-0.3 | 5-SPC | Dir-0.3 | 25-SPC | Dir-0.3 | 25-SPC |
| Ditto | $76.34 \pm 0.11$ | $65.17 \pm 0.17$ | $40.36 \pm 0.18$ | $44.83 \pm 0.19$ | $29.25 \pm 0.16$ | $36.58 \pm 0.18$ |
| FedRep | $76.49 \pm 0.15$ | $64.96 \pm 0.19$ | $41.54 \pm 0.16$ | $46.57 \pm 0.19$ | $31.22 \pm 0.15$ | $39.11 \pm 0.20$ |
| FedBABU | $78.22 \pm 0.14$ | $70.22 \pm 0.18$ | $44.18 \pm 0.15$ | $48.98 \pm 0.19$ | $32.91 \pm 0.14$ | $40.75 \pm 0.14$ |
| FedPAC | $76.53 \pm 0.13$ | $65.05 \pm 0.19$ | $41.60 \pm 0.16$ | $46.55 \pm 0.19$ | $31.20 \pm 0.17$ | $39.13 \pm 0.22$ |
| FedALA | $64.35 \pm 2.40$ | $55.58 \pm 1.88$ | $33.30 \pm 0.47$ | $36.41 \pm 0.58$ | $21.83 \pm 0.94$ | $27.83 \pm 1.59$ |
| FedAS | $78.69 \pm 0.17$ | $69.82 \pm 0.16$ | $45.65 \pm 0.18$ | $51.73 \pm 0.17$ | $36.06 \pm 0.13$ | $44.26 \pm 0.19$ |
| ConFREE | $76.73 \pm 0.16$ | $65.59 \pm 0.17$ | $41.91 \pm 0.16$ | $47.04 \pm 0.21$ | $31.57 \pm 0.15$ | $39.63 \pm 0.17$ |
| FedAvgFT | $79.08 \pm 0.11$ | $72.10 \pm 0.18$ | $46.55 \pm 0.15$ | $52.54 \pm 0.17$ | $36.83 \pm 0.17$ | $43.94 \pm 0.20$ |
| FedProxFT | $79.07 \pm 0.12$ | $72.07 \pm 0.18$ | $46.58 \pm 0.19$ | $52.49 \pm 0.18$ | $36.87 \pm 0.18$ | $43.96 \pm 0.19$ |
| FedSAMFT | $75.53 \pm 0.11$ | $66.30 \pm 0.16$ | $41.11 \pm 0.16$ | $44.89 \pm 0.16$ | $32.53 \pm 0.14$ | $40.25 \pm 0.16$ |
| `FedDRM` | $\mathbf{80.26 \pm 0.16}$ | $\mathbf{72.50 \pm 0.16}$ | $\mathbf{47.91 \pm 0.18}$ | $\mathbf{53.72 \pm 0.20}$ | $\mathbf{37.91 \pm 0.15}$ | $\mathbf{46.73 \pm 0.15}$ |

## 3.4 SENSITIVITY ANALYSIS

We evaluate the sensitivity of our method to several key factors. Experimental details are reported in App. F.2.

**Impact of weight $\lambda$ on system accuracy.** The reweighting parameter $\lambda$ is crucial for deploying the EL-based framework in the FL setting. As shown in Fig. 4, we observe the expected trade-off between two objectives: increasing $\lambda$ places more emphasis on image classification and less on client classification. This shift improves overall accuracy but reduces client accuracy, consistent with Thm. 2.5. The best balance between the two is achieved at $\lambda = 0.8$, where system accuracy peaks, marking the optimal trade-off for the task of guiding queries in the FL system.

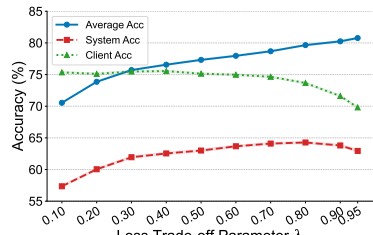

Figure 4: **Client & image accuracy trade-off** on CIFAR-10 under the Dir-0.3 setting.

**Covariate shift intensity.** We have already demonstrated in the main results that label shift is detrimental to all methods, with more severe shifts causing greater harm. To further examine the impact of covariate shift, we fix the degree of label shift and vary covariate shift at three intensity levels—low, mid, and high—by adjusting the parameters of the nonlinear color transformations. As shown in Fig. 5, the results reveal a clear trade-off: higher covariate shift intensifies differences between client data distributions, which facilitates client routing but simultaneously weakens information sharing across clients, thereby making image classification more difficult. Additional results examining the sensitivity of our method to the severity of label shift are provided in App. F.2.4.

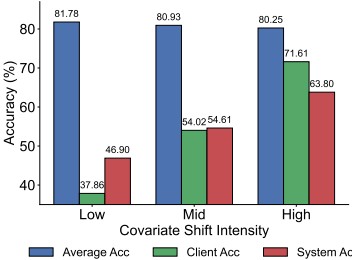

Figure 5: **Influence of covariate shift intensity** on CIFAR-10 under the Dir-0.3 setting.

**Backbone sharing strategy.** In our formulation, the target-class classification task uses the embedding $g_\theta(x)$ for an input feature $x$, while the client-classification task uses the embedding $h_\tau(g_\theta(x))$ for the same feature. Since both $g_\theta$ and $h_\tau$ are parameterized functions, the optimal sharing strategy between the two is not obvious. To explore this, we evaluate four cases: no sharing, shallow sharing, mid sharing, and deep sharing. As shown in Fig. 6, all strategies perform similarly, with shallow sharing slightly ahead. However, given the substantial increase in parameters for shallow sharing, deep sharing offers a more parameter-efficient alternative while maintaining strong performance.

**Number of clients.** To check scalability, we set the number of clients from 8 to 32 and compare `FedDRM` against the top-2 baselines from the main experiments. As shown in Tab. 3, while all methods exhibit a moderate performance decline as the client pool expands (a common challenge in FL), `FedDRM` consistently maintains a significant performance advantage across both system and average accuracy. This demonstrates that our method scales effectively, preserving its superiority even as the system grows.

Table 3: Sensitivity analysis on the number of clients $m$.

| Method | System Accuracy | | | | Average Accuracy | | | |
|---|---|---|---|---|---|---|---|---|
| | $m = 8$ | $m = 16$ | $m = 24$ | $m = 32$ | $m = 8$ | $m = 16$ | $m = 24$ | $m = 32$ |
| FedAS | $32.58 \pm 0.19$ | $36.55 \pm 0.21$ | $38.12 \pm 0.21$ | $34.50 \pm 0.18$ | $78.86 \pm 0.12$ | $73.28 \pm 0.15$ | $73.90 \pm 0.16$ | $73.45 \pm 0.15$ |
| FedAvgFT | $53.17 \pm 0.22$ | $50.61 \pm 0.20$ | $49.13 \pm 0.20$ | $45.07 \pm 0.21$ | $78.92 \pm 0.15$ | $73.41 \pm 0.16$ | $74.66 \pm 0.17$ | $74.18 \pm 0.15$ |
| FedDRM | $\mathbf{59.59 \pm 0.20}$ | $\mathbf{51.61 \pm 0.22}$ | $\mathbf{50.18 \pm 0.20}$ | $\mathbf{46.62 \pm 0.17}$ | $\mathbf{80.47 \pm 0.12}$ | $\mathbf{74.25 \pm 0.15}$ | $\mathbf{75.04 \pm 0.14}$ | $\mathbf{74.45 \pm 0.18}$ |

# 4 EXPERIMENT ON REAL MEDICAL DATASET

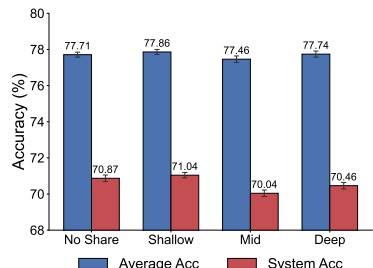

Figure 6: **Impact of the sharing strategy** on CIFAR-10 under the Dir-0.3 setting using LeNet (Lecun et al., 1998).

To further demonstrate FedDRM's effectiveness in healthcare, we evaluate it on the real medical dataset RETINA, following (Huang et al., 2025). RETINA comprises fundus images from three clinical centers—ACRIMA (Diaz-Pinto et al., 2019), Rim (Fumero Batista et al., 2020), and Refuge (Orlando et al., 2020). We exclude Drishti, which has only 82 images, while the others provide at least 385. Each $96 \times 96$ RGB image is labeled as Glaucomatous or Normal, creating a binary classification task.

This dataset naturally fits a 3-client FL system, with each client representing one center. The different image sources cause a covariate shift in RETINA. Furthermore, the class ratios (positive vs. negative) across the three datasets are $1.34$, $1.94$, and $0.46$, introducing a realistic label shift. Our experimental setup largely follows the CIFAR experiments, with several adjustments: the network embedding dimension is set to $4608$ and the projection dimension $512$. All methods train for $100$ communication rounds with a batch size of $32$. For fine-tuning-based methods, we allocate $90$ rounds to global training and $10$ to local fine-tuning.

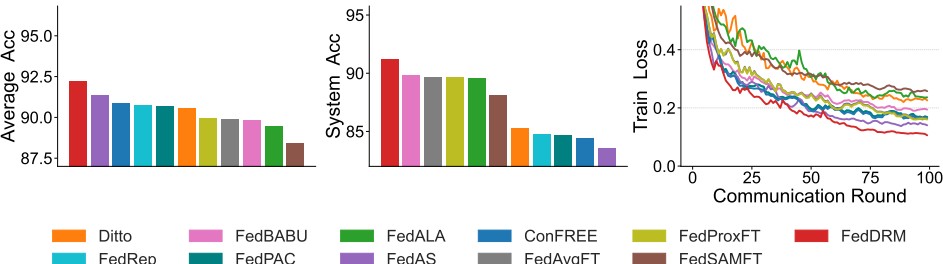

Figure 7: Average accuracy, system accuracy, and train loss on RETINA.

Fig. 7 shows that FedDRM consistently outperforms all baselines on RETINA. Measured in absolute accuracy points, FedDRM exceeds the competing methods by $0.83$–$3.77$ points in average accuracy and by $1.41$–$7.67$ points in system accuracy—substantial margins given the small size and pronounced heterogeneity of this dataset. These results underscore the robustness and practical relevance of FedDRM in the presence of simultaneous covariate and label shifts. Furthermore, FedDRM achieves the lowest training loss and the most stable convergence trajectory, demonstrating its effectiveness in capturing heterogeneous structure in real multi-center medical data.

# 5 CONCLUSION

This paper presents FedDRM, a novel FL paradigm that transforms statistical heterogeneity from a challenge into a resource. By introducing a unified EL based framework, FedDRM simultaneously learns accurate local models and a client-selection policy, enabling a central server to intelligently route new queries to the most appropriate client. Empirical results demonstrate that our method outperforms existing approaches in both client-level personalization and system-level utility, paving the way for more adaptive and resource-efficient FL systems that actively leverage statistical diversity. We believe that this work marks a meaningful step toward more adaptive, resource-efficient, and intelligent FL systems.

ACKNOWLEDGMENT

Zijian Wang & Qiong Zhang are supported by the National Key R&D Program of China Grant 2024YFA1015800 and the National Natural Science Foundation of China Grant 12301391. Xiaofei Zhang is supported by National Natural Science Foundation of China Grant 12501394 and the Natural Science Foundation of Hubei Province of China Grant 2025AFC035. Yukun Liu is supported by National Natural Science Foundation of China Grant 12571283. The contributions of Xin Zhang were made in a personal capacity and outside the scope of his employment at Meta. This research did not use any Meta data, tools, or internal resources. The views expressed in this paper are those of the authors and do not necessarily reflect the views of Meta.

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

## A    THE USE OF LARGE LANGUAGE MODELS (LLMS)

Large language models (LLMs) were used solely as assistive tools for language editing and polishing of the manuscript. The authors take full responsibility for the accuracy and integrity of the manuscript.

## B    DENSITY RATIO MODEL EXAMPLES

Many parametric distribution families including normal and Gamma are special cases of the DRM.

**Example B.1** (Normal distribution). For normal distribution $\phi(x; \mu, \sigma^2)$ with mean $\mu$ and variance $\sigma^2$. We have $\log\{\phi(x; \mu_1, \sigma_1^2)/\phi(x; \mu_2, \sigma_2^2)\} = \theta_0 + \theta_1 x + \theta_2 x^2$ where $\theta_0 = \log \sigma_2/\sigma_1 - (\mu_1^2/\sigma_1^2 - \mu_2^2/\sigma_2^2)/2$, $\theta_1 = \mu_1/\sigma_1^2 - \mu_2/\sigma_2^2$, $\theta_2 = (\sigma_2^{-2} - \sigma_1^{-2})/2$ and the basis function is $g(x) = (1, x, x^2)^\top$.

**Example B.2** (Gamma distribution). For gamma distribution with shape parameter $\alpha > 0$ and rate parameter $\beta > 0$. We have $\log\{f(x; \alpha_1, \beta_1)/f(x; \alpha_2, \beta_2)\} = \theta_0 + \theta_1 x + \theta_2 \log x$ where $\theta_0 = \log \Gamma(\alpha_2) - \log \Gamma(\alpha_1) + \alpha_1 \log \beta_1 - \alpha_2 \log \beta_2$, $\theta_1 = \beta_2 - \beta_1$, $\theta_2 = \alpha_1 - \alpha_2$ and the basis function is $g(x) = (1, x, \log x)^\top$.

## C    MATHEMATICAL DETAILS BEHIND FEDDRM

### C.1    DERIVATION OF (4)

*Proof.* By the total law of probability, the marginal density of $x$ is

$$
\begin{aligned}
dP_X^{(l)}(x) &= \sum_k \pi_{lk} d\mathbb{P}^{(l)}(x|y=k) \\
&= \sum_k \pi_{lk} \exp\{\alpha_{lk}^\dagger + \beta_k^\top g_\theta(x)\} d\mathbb{P}^{(l)}(x|y=1) \\
&= \sum_k \pi_{lk} \exp\{\alpha_{lk}^\dagger + \beta_k^\top g_\theta(x)\} \exp\{\gamma_l + \xi_l^\top h_\eta(g_\theta(x))\} d\mathbb{P}^{(0)}(x|y=1) \\
&= \sum_k \pi_{l1} \exp\{\alpha_k + \beta_k^\top g_\theta(x)\} \exp\{\gamma_l + \xi_l^\top h_\eta(g_\theta(x))\} d\mathbb{P}^{(0)}(x|y=1) \\
&= \sum_k \frac{\pi_{l1}}{\pi_{01}} \pi_{0k} \exp\{\alpha_{0k}^\dagger + \beta_k^\top g_\theta(x)\} \exp\{\gamma_l + \xi_l^\top h_\eta(g_\theta(x))\} d\mathbb{P}^{(0)}(x|y=1) \\
&= \frac{\pi_{l1}}{\pi_{01}} \exp\{\gamma_l + \xi_l^\top h_\eta(g_\theta(x))\} dP_X^{(0)}(x)
\end{aligned}
$$

Let $\gamma_l^\dagger = \gamma_l + \log(\pi_{l1}/\pi_{01})$ and divide $dP_X^{(0)}(x)$ on both sides completes the proof. □

### C.2    DERIVATION OF THE PROFILE LOG-LIKELIHOOD

Let $p = \{p_{ij}, j \in [n_i]\}_{i=1}^m$. Given $\zeta = (\alpha, \beta, \gamma, \xi, \theta, \eta)$, the empirical log-likelihood function as a function of $p$ becomes

$$
\ell_N(p) = \sum_{i=1}^m \sum_{j=1}^{n_i} \log p_{ij} + \text{constant}
$$

where the constant depends only on $\zeta$ and does not depend on $p$. We now maximize the empirical log-likelihood function with respect to $p$ under the constraint (5) using the Lagrange multiplier method.

Let

$$
\mathcal{L} = \sum_{i,j} \log p_{ij} - N\mu \sum_{i,j} p_{ij} - N \sum_{l=1}^m \rho_l \sum_{i,j} p_{ij}[\exp\{\gamma_l^\dagger + \xi_l^\top h_\eta(g_\theta(x_{ij}))\} - 1]
$$

Setting

$$
0 = \frac{\partial \mathcal{L}}{\partial p_{ij}} = \frac{1}{p_{ij}} - N\mu - N \sum_{l=1}^m \rho_l[\exp\{\gamma_l^\dagger + \xi_l^\top h_\eta(g_\theta(x_{ij}))\} - 1].
$$

Then multiply both sides by $p_{ij}$ and sum over $i$ and $j$, we have that

$$0 = \sum_{i,j} p_{ij} \frac{\partial \mathcal{L}}{\partial p_{ij}} = \sum_{i,j} \left\{ 1 - N\mu p_{ij} - \sum_{l=1}^{m} \rho_l p_{ij} [\exp\{\gamma_l^\dagger + \xi_l^\top h_\eta(g_\theta(x_{ij}))\} - 1] \right\}$$

$$= N - N\mu$$

this gives $\mu = 1$. Hence, we get

$$p_{ij} = \frac{1}{N \left\{ 1 + \sum_l \rho_l [\exp\{\gamma_l^\dagger + \xi_l^\top h_\eta(g_\theta(x_{ij}))\} - 1] \right\}}$$

where $\rho_l$s are solutions to

$$\sum_{i,j} \frac{\exp\{\gamma_l^\dagger + \xi_l^\top h_\eta(g_\theta(x_{ij}))\} - 1}{1 + \sum_{l'} \rho_{l'} [\exp\{\gamma_{l'}^\dagger + \xi_{l'}^\top h_\eta(g_\theta(x_{ij}))\} - 1]} = 0$$

by plugin the expression for $p_{ij}$ into the second constraints (5).

### C.3  DERIVATION OF THE VALUE OF LAGRANGE MULTIPLIER AT OPTIMAL

Recall that the profile log-EL has the following form

$$p\ell_N(\boldsymbol{\zeta}) = \sum_{i,j,k} \mathbb{1}(y_{ij} = k) \log \mathbb{P}(y_{ij} = k | x_{ij}) + \sum_{i,j} \{\gamma_i^\dagger + \xi_i^\top h_\eta(g_\theta(x_{ij})) + \log p_{ij}(\boldsymbol{\zeta})\}$$

$$= \sum_{i,j,k} \mathbb{1}(y_{ij} = k) \log \mathbb{P}(y_{ij} = k | x_{ij}) + \sum_{i,j} \left\{ \gamma_i^\dagger + \xi_i^\top h_\eta(g_\theta(x_{ij})) \right\}$$

$$- \sum_{i,j} \log \left\{ 1 + \sum_{l=1}^{m} \rho_l \left[ \exp\{\gamma_l^\dagger + \xi_l^\top h_\eta(g_\theta(x_{ij}))\} - 1 \right] \right\}$$

where $\rho_l$s are solutions to

$$\sum_{i,j} \frac{\exp\{\gamma_l^\dagger + \xi_l^\top h_\eta(g_\theta(x_{ij}))\} - 1}{1 + \sum_{l'} \rho_{l'} [\exp\{\gamma_{l'}^\dagger + \xi_{l'}^\top h_\eta(g_\theta(x_{ij}))\} - 1]} = 0.$$

Taking the partial derivative with respect to $\gamma_l^\dagger$, we have

$$0 = \frac{\partial p\ell_N}{\partial \gamma_l^\dagger} = n_l - \sum_{i,j} \frac{\rho_l \exp\{\gamma_l^\dagger + \xi_l^\top h(\boldsymbol{x}_{ij})\}}{1 + \sum_{l'} \rho_{l'} [\exp\{\gamma_{l'}^\dagger + \xi_{l'}^\top h_\eta(g_\theta(x_{ij}))\} - 1]}$$

$$+ \sum_{i,j} \frac{\sum_{l'} (\partial \rho_{l'} / \partial \gamma_l^\dagger) \left[ \exp(\gamma_{l'}^\dagger + \xi_{l'}^\top h_\eta(g_\theta(\boldsymbol{x}_{ij}))) - 1 \right]}{1 + \sum_{l'} \rho_{l'} [\exp\{\gamma_{l'}^\dagger + \xi_{l'}^\top h_\eta(g_\theta(x_{ij}))\} - 1]}$$

$$= n_l - N\rho_l \sum_{i,j} p_{ij} \exp\{\gamma_l^\dagger + \xi_l^\top h(\boldsymbol{x}_{ij})\} + N \sum_{l'} \frac{\partial \rho_{l'}}{\partial \gamma_l^\dagger} \sum_{i,j} p_{ij} [\exp\{\gamma_l^\dagger + \xi_l^\top h(\boldsymbol{x}_{ij})\} - 1]$$

$$= n_l - N\rho_l.$$

The last inequality is based on the constraint in (5). Hence, we have $\rho_l = n_l / N$ which completes the proof.

### C.4 DUAL FORM OF THE PROFILE LOG EL

At the optimal value, we have $\rho_l = n_l/N$ with $N = \sum_{l=1}^{m} n_l$. Plugin this value into the profile log-EL, we then get

$$
\begin{aligned}
p\ell_N(\zeta) &= \sum_{i,j} \log \left\{ \frac{\exp(\alpha_{y_{ij}} + \beta_{y_{ij}}^\top g_\theta(x_{ij}))}{\sum_j \exp(\alpha_j + \beta_j^\top g_\theta(x_{ij}))} \right\} + \sum_{i,j} \log \left\{ \frac{\exp\{\gamma_i^\dagger + \xi_i^\top h_\eta(g_\theta(x_{ij}))\}}{\sum_{l=1}^{m} \frac{n_l}{N} \exp\{\gamma_l^\dagger + \xi_l^\top h_\eta(g_\theta(x_{ij}))\}} \right\} \\
&= \sum_{i,j} \log \left\{ \frac{\exp(\alpha_{y_{ij}} + \beta_{y_{ij}}^\top g_\theta(x_{ij}))}{\sum_j \exp(\alpha_j + \beta_j^\top g_\theta(x_{ij}))} \right\} + \sum_{i,j} \log \left\{ \frac{(n_1/n_i)\exp\{\gamma_i^\ddagger + \xi_i^\top h_\eta(g_\theta(x_{ij}))\}}{\sum_{l=1}^{m}(\frac{n_1}{N})\exp\{\gamma_l^\ddagger + \xi_l^\top h_\eta(g_\theta(x_{ij}))\}} \right\} \\
&= \sum_{i,j} \log \left\{ \frac{\exp(\alpha_{y_{ij}} + \beta_{y_{ij}}^\top g_\theta(x_{ij}))}{\sum_j \exp(\alpha_j + \beta_j^\top g_\theta(x_{ij}))} \right\} + \sum_{i,j} \log \left\{ \frac{(\exp\{\gamma_i^\ddagger + \xi_i^\top h_\eta(g_\theta(x_{ij}))\}}{\sum_{l=1}^{m} \exp\{\gamma_l^\ddagger + \xi_l^\top h_\eta(g_\theta(x_{ij}))\}} \right\} \\
&\quad - \sum_{i,j} \log \left( \frac{n_i}{N} \right).
\end{aligned}
$$

The last term is an additive constant; the maximization does not depend on its value, which completes the proof.

## D  GENERALIZATION TO OTHER TYPES OF DATA HETEROGENEITY

In this section, we detail how our method generalizes to the setting where both $Y|X$ and $X$ differ across clients. Recall that the log empirical likelihood function is

$$
\begin{aligned}
\ell_N(p, \zeta) &= \sum_{i=1}^{m} \sum_{j=1}^{n_i} \log P_{X,Y}^{(i)}(\{X_{ij}, Y_{ij}\}) \\
&= \sum_{i,j,k} \mathbb{1}(Y_{ij} = k) \log \mathbb{P}^{(i)}(Y = k|X_{ij}) + \sum_{i,j} \log P_X^{(i)}(\{X_{ij}\})
\end{aligned}
$$

We assume that each client has its own linear head for the conditional distribution:

$$
\mathbb{P}^{(i)}(Y = k|X = x) = \frac{\exp(\alpha_{ik} + \beta_{ik}^\top g_\theta(x))}{\sum_j \exp(\alpha_{ikj} + \beta_{ij}^\top g_\theta(x))},
$$

The marginal distributions $P_X^{(i)}$ are linked as in Theorem 2.1:

$$
\frac{dP_X^{(i)}}{dP_X^{(0)}}(x) = \exp\{\gamma_i^\dagger + \xi_i^\top h_\tau(g_\theta(x))\}
$$

where $P_X^{(0)}$ is an unspecified reference measure. Using a non-parametric reference distribution, we set

$$
p_{ij} = P_X^{(0)}(\{X_{ij}\}) \geq 0, \quad \forall i \in [m], \ j \in [n_i],
$$

subject to the constraints

$$
\sum_{i=1}^{m} \sum_{j=1}^{n_i} p_{ij} = 1, \quad \sum_{i=1}^{m} \sum_{j=1}^{n_i} p_{ij} \exp\{\gamma_l^\dagger + \xi_l^\top h_\tau(g_\theta(X_{ij}))\} = 1, \quad \forall l \in [m].
$$

Then, the log empirical likelihood across clients is

$$
\ell_N(p, \zeta) = \sum_{i,j,k} \mathbb{1}(Y_{ij} = k) \log \mathbb{P}^{(i)}(Y = k|X_{ij}) + \sum_{i,j} \{\gamma_i^\dagger + \xi_i^\top h_\tau(g_\theta(X_{ij})) + \log p_{ij}\}.
$$

The profile log-EL of $\zeta$ is defined as

$$
p\ell_N(\zeta) = \sup_p \ell_N(p, \zeta)
$$

where the supremum is taken under the constraints above. Applying the method of Lagrange multipliers, we obtain the analytical form

$$p\ell_N(\zeta) = \sum_{i,j,k} \mathbb{1}(Y_{ij} = k) \log \mathbb{P}^{(i)}(Y_{ij} = k | X_{ij}) + \sum_{i,j} \{\gamma_i^\dagger + \xi_i^\top h_\tau(g_\theta(x_{ij})) + \log p_{ij}(\zeta)\}$$

where

$$p_{ij}(\zeta) = N^{-1} \Big\{ 1 + \sum_{l=1}^m \rho_l \Big[ \exp\{\gamma_l^\dagger + \xi_l^\top h_\tau(g_\theta(x_{ij}))\} - 1 \Big] \Big\}^{-1}$$

and the Lagrange multipliers $\{\rho_l\}_{l=1}^m$ solves

$$\sum_{i,j} \frac{\exp\{\gamma_l^\dagger + \xi_l^\top h_\tau(g_\theta(x_{ij}))\} - 1}{\sum_{l'} \rho_{l'} \Big[ \exp(\gamma_{l'}^\dagger + \xi_{l'}^\top h_\tau(g_\theta(x_{ij}))) - 1 \Big]} = 0.$$

Using the dual argument from Appendix C.3, the profile log-EL can be rewritten as

$$
\begin{aligned}
p\ell_N(\zeta) &= \sum_{i,j} \log \left\{ \frac{\exp(\alpha_{i,y_{ij}} + \beta_{i,y_{ij}}^\top g_\theta(x_{ij}))}{\sum_j \exp(\alpha_{ij} + \beta_{ij}^\top g_\theta(x_{ij}))} \right\} + \sum_{i,j} \log \left\{ \frac{\exp\{\gamma_i^\dagger + \xi_i^\top h_\eta(g_\theta(x_{ij}))\}}{\sum_{l=1}^m \frac{n_l}{N} \exp\{\gamma_l^\dagger + \xi_l^\top h_\eta(g_\theta(x_{ij}))\}} \right\} \\
&= \sum_{i,j} \log \left\{ \frac{\exp(\alpha_{i,y_{ij}} + \beta_{i,y_{ij}}^\top g_\theta(x_{ij}))}{\sum_j \exp(\alpha_{ij} + \beta_{ij}^\top g_\theta(x_{ij}))} \right\} + \sum_{i,j} \log \left\{ \frac{(n_1/n_i) \exp\{\gamma_i^\ddagger + \xi_i^\top h_\eta(g_\theta(x_{ij}))\}}{\sum_{l=1}^m (\frac{n_1}{N}) \exp\{\gamma_l^\ddagger + \xi_l^\top h_\eta(g_\theta(x_{ij}))\}} \right\} \\
&= \sum_{i,j} \log \left\{ \frac{\exp(\alpha_{i,y_{ij}} + \beta_{i,y_{ij}}^\top g_\theta(x_{ij}))}{\sum_j \exp(\alpha_{ij} + \beta_{ij}^\top g_\theta(x_{ij}))} \right\} + \sum_{i,j} \log \left\{ \frac{(\exp\{\gamma_i^\ddagger + \xi_i^\top h_\eta(g_\theta(x_{ij}))\}}{\sum_{l=1}^m \exp\{\gamma_l^\ddagger + \xi_l^\top h_\eta(g_\theta(x_{ij}))\}} \right\} \\
&\quad - \sum_{i,j} \log \left( \frac{n_i}{N} \right).
\end{aligned}
$$

As a result, the loss function remains additive in two cross-entropy terms corresponding to different tasks. The key difference from the covariate shift case is that, for the client-classification task, each client now has its own linear head.

# E   SYSTEM ACCURACY & CONVERGENCE RATE TRADE-OFF

We show the proof of Theorem 2.5 in this section. To simplify the notation, we consider the following loss: To assure strong convexity, we minimize the following objective function:

$$\ell_i^\rho(\zeta) = \underbrace{\frac{\rho}{2}\|\gamma\|_2^2 - \frac{1-\lambda}{n_i} \sum_{j=1}^{n_i} \log \frac{\exp(\gamma_i^\top z_{ij})}{\sum_{q=1}^m \exp(\gamma_q^\top z_{ij})}}_{=:\ell_i^{\rho,\text{client}}(\gamma)} + \underbrace{\frac{\rho}{2}\|\beta\|_2^2 - \frac{\lambda}{n_i} \sum_{j=1}^{n_i} \log \frac{\exp(\beta_{y_{ij}}^\top x_{ij})}{\sum_{k=1}^K \exp(\beta_k^\top x_{ij})}}_{=:\ell_i^{\rho,\text{class}}(\beta)},$$

Then, where $\zeta = (\gamma, \beta)$ stacks the parameters for client-classification $\gamma$ and task-classification $\beta$. the global objective is $\ell^\rho(\zeta) = m^{-1} \sum_{i=1}^m \ell_i^\rho(\zeta)$.

**Problem setting:** Assume the data is generated according to the true multinomial logistic model with parameters $\zeta^{\text{true}} = (\gamma^{\text{true}}, \beta^{\text{true}})$. i.e.,

$$\Pr(Y_{\text{client}} = q | z) = \frac{\exp(\gamma_q^{*\top} z)}{\sum_{r=1}^m \exp(\gamma_r^{*\top} z)}, \qquad \Pr(Y_{\text{class}} = k | x) = \frac{\exp(\beta_k^{*\top} x)}{\sum_{r=1}^K \exp(\beta_r^{*\top} x)}.$$

Let $\widehat{\zeta}_N$ denote the minimizer of $\ell^\rho(\zeta)$ with $N = \sum_{i=1}^m n_i$ as the total number of samples.

**Total error.** Let $\zeta^T$ be the output of the algorithm after $T$ steps. We decompose the error using the triangle inequality as follows:

$$\|\zeta^T - \zeta^{\text{true}}\|_2 \leq \underbrace{\|\zeta^T - \widehat{\zeta}_N\|_2}_{\text{optimization error}} + \underbrace{\|\widehat{\zeta}_N - \zeta^{\text{true}}\|_2}_{\text{statistical error}}. \tag{8}$$

We know bound these two terms respectively.

**Lemma E.1** (Asymptotic normality). *As $N \to \infty$, the estimator $\widehat{\zeta}_N$ satisfies*

$$\sqrt{N}\,(\widehat{\zeta}_N - \zeta^{true}) \;\xrightarrow{d}\; \mathcal{N}\Big(0, \mathcal{I}(\zeta^{true})^{-1}\Big),$$

*where the Fisher information is block diagonal:*

$$\mathcal{I}(\zeta^{true}) = \begin{bmatrix} (1-\lambda)\,\mathcal{I}_\gamma + \rho I & 0 \\ 0 & \lambda\,\mathcal{I}_\beta + \rho I \end{bmatrix},$$

*with*

$$\mathcal{I}_\gamma = \mathbb{E}\left\{\big(\mathrm{diag}(p_\gamma(z)) - p_\gamma(z)p_\gamma(z)^\top\big) \otimes (zz^\top)\right\}, \;\; \mathcal{I}_\beta = \mathbb{E}\left\{\big(\mathrm{diag}(p_\beta(x)) - p_\beta(x)p_\beta(x)^\top\big) \otimes (xx^\top)\right\},$$

*where $p_\beta(x) = (\exp(\beta_1^\top x)/\sum_j \exp(\beta_j^\top x), \ldots, \exp(\beta_{dim(\beta)}^\top x)/\sum_j \exp(\beta_j^\top x))^\top$, and $I$ is the identity matrix.*

*Proof.* This result follows from the well-established asymptotic properties of maximum likelihood estimators (Van der Vaart, 2000, Section 5.5). □

**Statistical error**. By Lemma E.1, we have

$$N\|\widehat{\gamma}_N - \gamma^{\text{true}}\|^2 = O_p\Big(\frac{d}{(1-\lambda)\|\mathcal{I}_\gamma\|_{\min} + \rho}\Big), \qquad N\|\widehat{\beta}_N - \beta^{\text{true}}\|^2 = O_p\Big(\frac{p}{\lambda\|\mathcal{I}_\beta\|_{\min} + \rho}\Big), \quad (9)$$

where $\|A\|_{\min} = \lambda_{\min}(A)$ is the operator norm, $d$ and $p$ are dimensions of $z$ and $x$ respectively.

**Optimization error.** For communication round $t = 0, 1, 2, \ldots, T-1$, the server holds $\zeta^t$ and each client $i$ sets $\zeta_{i,0}^t = \zeta^t$ and performs $E$ local gradient steps:

$$\zeta_{i,r+1}^t = \zeta_{i,r}^t - \eta\nabla\ell_i^\rho(\zeta_{i,r}^t), \qquad r = 0, \ldots, E-1.$$

After $E$ steps each client returns $\zeta_{i,E}^t$ and the server aggregates $\zeta^{t+1} = m^{-1}\sum_{i=1}^m \zeta_{i,E}^t$.

Define $G^2(\zeta) = m^{-1}\sum_{i=1}^m \|\nabla\ell_i^\rho(\zeta) - \nabla\ell^\rho(\zeta)\|_2^2$. It can be decomposed nicely as $G^2(\zeta) = (1-\lambda)^2 G_{\text{client}}^2(\gamma) + \lambda^2 G_{\text{class}}^2(\beta)$. Let $\bar{G}^2$, $\bar{G}_{\text{client}}^2$, and $\bar{G}_{\text{class}}^2$ denote the corresponding maximum values across updating rounds $t = 0, 2, \ldots, T-1$. Then, $\bar{G}^2 \le (1-\lambda)^2 \bar{G}_{\text{client}}^2 + \lambda^2 \bar{G}_{\text{class}}^2$.

In the convergence proof below, we omit the subscript $\rho$ since it does not influence the convergence rate. Because $\ell$ is $\mu$-strongly convex and $L$-smooth, a single full-gradient step satisfies

$$\begin{aligned} \|x - \eta\nabla\ell(x) - \widehat{\zeta}_N\|_2^2 &= \|x - \widehat{\zeta}_N\|_2^2 - 2\eta\langle\nabla\ell(x), x - \widehat{\zeta}_N\rangle + \eta^2\|\nabla\ell(x)\|_2^2 \\ &\le \|x - \widehat{\zeta}_N\|_2^2 - 2\eta\mu\|x - \widehat{\zeta}_N\|_2^2 + \eta^2 L^2\|x - \widehat{\zeta}_N\|_2^2 \\ &\le (1 - \eta\mu)\|x - \widehat{\zeta}_N\|_2^2. \end{aligned}$$

For client $i$ at local step $r$:

$$\begin{aligned} \|\zeta_{i,r+1}^t - \widehat{\zeta}_N\|_2^2 &= \|(\zeta_{i,r}^t - \eta\nabla\ell(\zeta_{i,r}^t) - \widehat{\zeta}_N) + \eta(\nabla\ell(\zeta_{i,r}^t) - \nabla\ell_i(\zeta_{i,r}^t))\|^2 \\ &\le (1 - \eta\mu)\|\zeta_{i,r}^t - \widehat{\zeta}_N\|_2^2 + \eta^2\|\nabla\ell(\zeta_{i,r}^t) - \nabla\ell_i(\zeta_{i,r}^t)\|^2 \end{aligned}$$

Iterating over $E$ local steps gives

$$\|\zeta_{i,E}^t - \widehat{\zeta}_N\|_2^2 \le (1-\eta\mu)^E\|\zeta^t - \widehat{\zeta}_N\|_2^2 + \eta^2\sum_{r=0}^{E-1}(1-\eta\mu)^{E-1-r}\|\nabla\ell_i(\zeta_{i,r}^t) - \nabla\ell(\zeta_{i,r}^t)\|_2^2.$$

Averaging over $i = 1, \ldots, m$ and using convexity of squared norm:

$$\|\zeta^{t+1} - \widehat{\zeta}_N\|_2^2 \le (1-\eta\mu)^E\|\zeta^t - \widehat{\zeta}_N\|_2^2 + \eta^2\sum_{r=0}^{E-1}\frac{1}{m}\sum_{i=1}^m\|\nabla\ell_i(\zeta_{i,r}^t) - \nabla\ell(\zeta_{i,r}^t)\|_2^2.$$

Using $L$-smoothness and $\eta LE \leq 1/4$, one can show (via induction on $r$ and triangle inequalities)

$$\frac{1}{m}\sum_{i=1}^{m}\|\nabla\ell_i(\zeta_{i,r}^t) - \nabla\ell(\zeta_{i,r}^t)\|_2^2 \leq \bar{G}^2,$$

where $\bar{G}^2$ is the heterogeneity measure. Summing over $r = 0, \ldots, E-1$ gives

$$\eta^2\sum_{r=0}^{E-1}\frac{1}{m}\sum_{i=1}^{m}\|\nabla\ell_i(\zeta_{i,r}^t) - \nabla\ell(\zeta_{i,r}^t)\|_2^2 \leq \eta^2 E^2 \bar{G}^2.$$

Combine the above:

$$\|\zeta^{t+1} - \widehat{\zeta}_N\|_2^2 \leq (1-\eta\mu)^E\|\zeta^t - \widehat{\zeta}_N\|_2^2 + \eta^2 E^2 \bar{G}^2.$$

Let $s_t := \|\zeta^t - \widehat{\zeta}_N\|_2^2$ and $\alpha := (1-\eta\mu)^E$, $B := \eta^2 E^2 \bar{G}^2$. Then

$$s_{t+1} \leq \alpha s_t + B \quad \Rightarrow \quad s_T \leq \alpha^T s_0 + B\sum_{j=0}^{T-1}\alpha^j = \alpha^T s_0 + \frac{B(1-\alpha^T)}{1-\alpha} \leq \alpha^T s_0 + \frac{B}{1-\alpha}.$$

This yields the desired bound

$$\|\zeta^T - \widehat{\zeta}_N\|_2^2 \leq (1-\eta\mu)^{ET}\|\zeta^0 - \widehat{\zeta}_N\|_2^2 + \frac{\eta^2 E^2 \bar{G}^2}{1-(1-\eta\mu)^E}. \tag{10}$$

Since $1 - (1-\eta\mu)^E \geq 1 - e^{-\eta\mu E} \geq \frac{1}{2}\min\{1, \eta\mu E\}$, the steady-state error is of order $O(\eta^2 E^2 \bar{G}^2/(\eta\mu E)) = O(\eta E \bar{G}^2/\mu)$, i.e., FedAvg converges linearly to a neighborhood of radius proportional to $\sqrt{\eta E \bar{G}^2/\mu}$.

Combining (9) and (10) with (8) gives the final result that

$$\|\zeta^T - \zeta^{\text{true}}\|^2 = O_p\left(\frac{\{(1-\lambda)\|\mathcal{I}_\gamma\|_{\min} + \rho\}^{-1} + \{\lambda\|\mathcal{I}_\beta\|_{\min} + \rho\}^{-1}}{N} + \frac{\eta^2 E^2 \bar{G}^2}{1-(1-\eta\mu)^E}\right),$$

as both $T, N \to \infty$, This along with $\bar{G}^2 \leq (1-\lambda)^2\bar{G}_{\text{client}}^2 + \lambda^2\bar{G}_{\text{class}}^2$ completes the proof of the theorem.

# F EXPERIMENT DETAILS

## F.1 VISUALIZATION OF COVARIATE SHIFT AND LABEL SHIFT

In the main experiment, we simulate covariate shift by applying three distinct nonlinear transformations to each client's dataset. Specifically, we use gamma correction with $\gamma \in \{0.6, 1.4\}$, hue adjustment with $\Delta h \in \{-0.1, 0.1\}$, and saturation scaling with $\kappa \in \{0.5, 1.5\}$. This creates $2^3 = 8$ unique combinations of transformations, corresponding to an 8-client setting where each client possesses a visually distinct data distribution. A visualization of a single image sampled from CIFAR-10 after applying these transformations is shown in Fig. 8. As can be clearly seen, the resulting differences in feature distributions across clients are visually striking, highlighting the significant covariate shift simulated in our experiments. We visualize the two types of label shift used in the main experiment in Fig. 9. The figures show the number of samples from each class across 8 clients. As observed, the 5-SPC setting assigns at most 5 classes to each client, whereas the Dir-0.3 setting distributes more classes per client. Thus, the label shift under Dir-0.3 is less severe than under 5-SPC. Our experimental results confirm this observation: all methods achieve higher performance under the less severe Dir-0.3 case.

## F.2 SENSITIVITY ANALYSIS DETAILS

For all subsequent sensitivity analyses, unless otherwise specified, we use CIFAR-10 under the Dir-0.3 setting. The details of each experiment are provided below.

Covariate Shift - Main Experiment

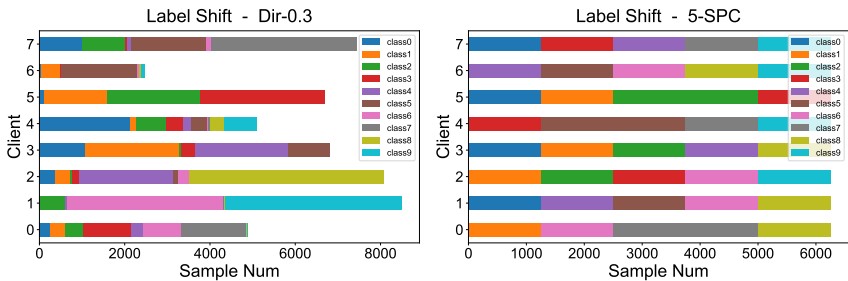

Figure 8: Visualization of a sample from CIFAR-10 under various nonlinear transformations.

Figure 9: Visualization of client data distribution on CIFAR-10 under Dir-0.3 and 5-SPC settings.

### F.2.1 NUMBER OF CLIENTS

To investigate the impact of the number of clients, we adopt a more fine-grained strategy for simulating covariate shift. We expand the parameter space for each nonlinear transformation to three distinct values: gamma correction with $\gamma \in \{0.6, 1.0, 1.4\}$, hue adjustment with $\Delta h \in \{-0.15, 0.0, 0.15\}$, and saturation scaling with $\kappa \in \{0.4, 1.0, 1.6\}$. Furthermore, we introduce an additional binary transformation, posterization, which reduces the number of bits for each color channel to create a flattening effect on the image's color palette. A visualization of these transformations is presented in Fig. 10. In the $n$-client setting, we apply the first $n$ transformations from this pool.

In our experiments, we set the maximum number of clients to 32. This is due to two primary challenges. First, as the number of clients increases, the amount of data partitioned to each client diminishes significantly. This data scarcity creates a scenario where fine-tuning-based methods gain an inherent advantage, as each client's local train and test distributions are identical. Second, it is hard to design a simulation strategy for covariate shift that is both sufficiently distinct and aligned with the model's inductive bias when the number of clients becomes very large.

### F.2.2 COVARIATE SHIFT INTENSITY

To evaluate the robustness of our method under varying degrees of covariate shift, we construct three intensity levels—low, mid, and high—by adjusting the parameter ranges of the nonlinear transformations. The specific value ranges for each level are detailed as follows: (1) **Low**: $\gamma \in \{0.9, 1.1\}, \Delta h \in \{-0.01, 0.01\}, \kappa \in \{0.9, 1.1\}$. (2) **Mid**: $\gamma \in \{0.75, 1.25\}, \Delta h \in \{-0.05, 0.05\}, \kappa \in \{0.7, 1.3\}$. (3) **High**: $\gamma \in \{0.6, 1.4\}, \Delta h \in \{-0.1, 0.1\}, \kappa \in \{0.5, 1.5\}$. Visualizations corresponding to these levels are presented in Fig. 11, Fig. 12, and Fig. 8. It can be seen that the induced covariate shift is nearly imperceptible at the low level and escalates to a stark distinction at the high level, clearly illustrating the progressive intensity of the shift.

Covariate Shift - Sensitivity to Number of Clients

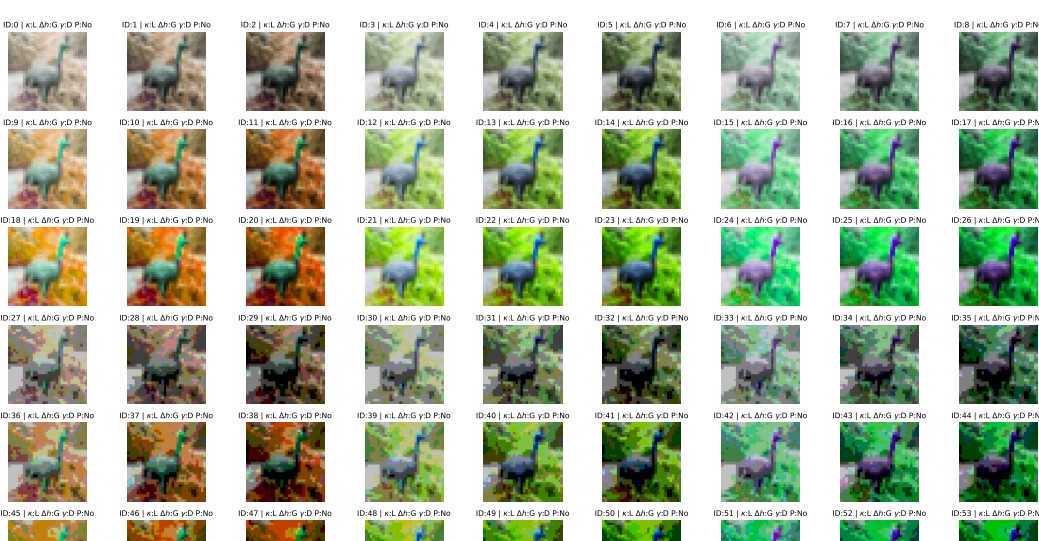

Figure 10: Visualization of a CIFAR-10 sample under covariate shift with a larger number of clients.

Covariate Shift - Low Level

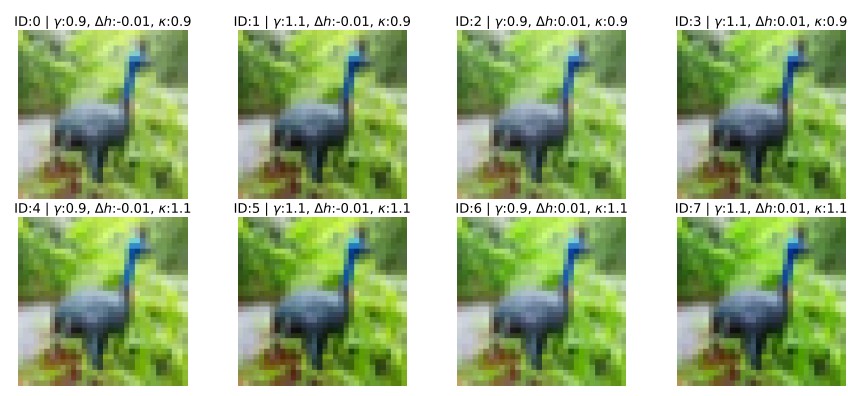

Figure 11: Visualization of a sample from CIFAR-10 under low covariate shift intensity.

### F.2.3 BACKBONE SHARING STRATEGY

In our formulation, the target-class classification task uses the embedding $g_\theta(x)$ for an input feature $x$, while the client-classification task uses the embedding $h_\tau(g_\theta(x))$ for the same feature. Since both $g_\theta$ and $h_\tau$ are parameterized functions, the optimal sharing strategy between the two is not obvious. To explore this, we investigate four backbone-sharing strategies based on LeNet: no sharing, shallow sharing, mid sharing, and deep sharing. The network architectures are illustrated in Fig. 13. From (a) to (c), the discrepancy between the embeddings for the two tasks decreases, while the number of learnable parameters also reduces. Our empirical results in Fig. 6 show that all strategies perform similarly, with shallow sharing slightly ahead. However, given the substantial increase in parameters for shallow sharing, deep sharing offers a more parameter-efficient alternative while maintaining strong performance.

Covariate Shift - Mid Level

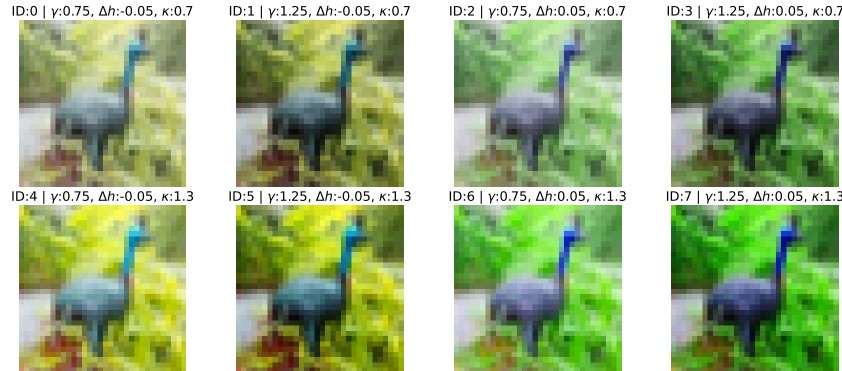

Figure 12: Visualization of a sample from CIFAR-10 under mid covariate shift intensity.

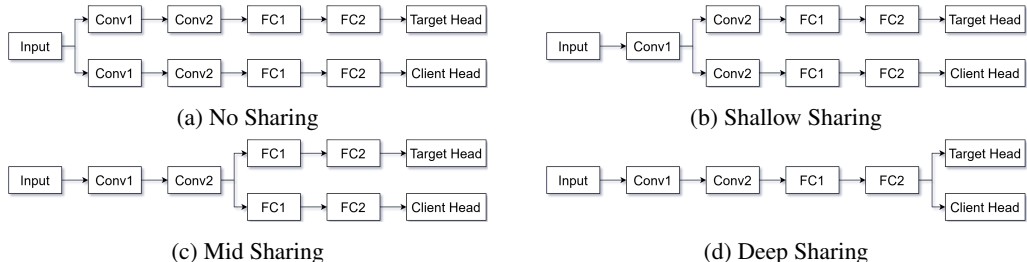

Figure 13: Visualization of the four different parameter sharing strategies.

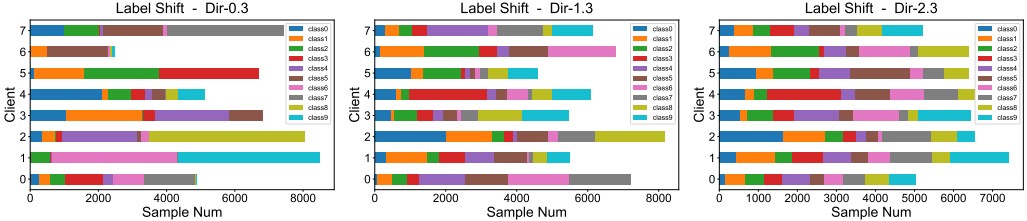

Figure 14: Visualization of client data distribution under Dir-$0.3/1.3/2.3$ settings.

Table 4: System accuracy and average accuracy under different Dirichlet parameter $\alpha$ values.

| Method | System Accuracy | | | Average Accuracy | | |
| --- | --- | --- | --- | --- | --- | --- |
| | $\alpha = 0.3$ | $\alpha = 1.3$ | $\alpha = 2.3$ | $\alpha = 0.3$ | $\alpha = 1.3$ | $\alpha = 2.3$ |
| Ditto | $47.64 \pm 0.25$ | $43.78 \pm 0.23$ | $44.04 \pm 0.20$ | $76.34 \pm 0.11$ | $55.56 \pm 0.18$ | $52.76 \pm 0.18$ |
| FedRep | $24.96 \pm 0.19$ | $35.45 \pm 0.22$ | $37.87 \pm 0.22$ | $76.49 \pm 0.15$ | $55.09 \pm 0.17$ | $52.95 \pm 0.16$ |
| FedBABU | $57.43 \pm 0.17$ | $54.60 \pm 0.19$ | $54.25 \pm 0.20$ | $78.22 \pm 0.14$ | $61.61 \pm 0.16$ | $59.83 \pm 0.20$ |
| FedPAC | $25.14 \pm 0.21$ | $35.53 \pm 0.21$ | $37.84 \pm 0.20$ | $76.53 \pm 0.13$ | $55.08 \pm 0.15$ | $52.95 \pm 0.16$ |
| FedALA | $61.33 \pm 0.17$ | $48.47 \pm 0.18$ | $46.68 \pm 0.20$ | $64.35 \pm 2.40$ | $49.14 \pm 0.62$ | $47.29 \pm 0.50$ |
| FedAS | $28.76 \pm 0.19$ | $41.43 \pm 0.20$ | $46.08 \pm 0.22$ | $78.69 \pm 0.17$ | $59.83 \pm 0.19$ | $58.17 \pm 0.18$ |
| ConFREE | $25.66 \pm 0.22$ | $36.12 \pm 0.21$ | $38.78 \pm 0.23$ | $76.73 \pm 0.16$ | $55.56 \pm 0.18$ | $53.58 \pm 0.15$ |
| FedAvgFT | $54.90 \pm 0.22$ | $54.80 \pm 0.18$ | $55.61 \pm 0.19$ | $79.08 \pm 0.11$ | $62.87 \pm 0.16$ | $61.94 \pm 0.17$ |
| FedProxFT | $55.01 \pm 0.20$ | $54.84 \pm 0.15$ | $55.62 \pm 0.18$ | $79.07 \pm 0.12$ | $62.84 \pm 0.15$ | $61.86 \pm 0.19$ |
| FedSAMFT | $55.83 \pm 0.21$ | $49.15 \pm 0.17$ | $47.71 \pm 0.16$ | $75.53 \pm 0.11$ | $55.46 \pm 0.19$ | $53.54 \pm 0.15$ |
| FedDRM | $\mathbf{63.85 \pm 0.18}$ | $\mathbf{56.83 \pm 0.18}$ | $\mathbf{56.00 \pm 0.23}$ | $\mathbf{80.25 \pm 0.14}$ | $\mathbf{64.04 \pm 0.16}$ | $\mathbf{62.30 \pm 0.15}$ |

### F.2.4 LABEL SHIFT INTENSITY

To evaluate the robustness of our method under varying degrees of label shift, we compare our method with the baselines across a range of Dirichlet parameters $\alpha \in \{0.3, 1.3, 2.3\}$. Visualizations corresponding to these settings are presented in Fig. 14. The corresponding results for system accuracy and average accuracy are presented in Tab. 4.

Consistent with prior work (Xu et al., 2023), we can see that smaller $\alpha$ values—corresponding to higher data heterogeneity—lead to higher average accuracy for all methods. This occurs because each client's training and testing data are drawn from the same distribution. As $\alpha$ decreases, the local label distributions become increasingly skewed, with some classes receiving negligible probability mass. This effectively reduces the number of classes present on each client, thereby simplifying the local classification problem relative to the balanced case. For system accuracy, we find that this trend persists for our method, as the only additional component is the client-routing step, which does not alter the underlying behavior of local classification. In contrast, methods such as FedRep exhibit increasing system accuracy as $\alpha$ grows (i.e., as the label distributions become more homogeneous). When $\alpha$ is small, the local models become highly personalized and fail to reach a consistent consensus across clients, causing majority voting to misroute queries and thus lowering system accuracy. As $\alpha$ increases, this inconsistency diminishes, and the aggregated routing accuracy improves. These results further confirm that our method remains robust across varying degrees of label shift.

