# OpenReview forum: "Beyond Aggregation: Guiding Clients in Heterogeneous Federated Learning"
_ICLR.cc/2026/Conference — ICLR 2026 Poster_

### Official Review · Reviewer_C2JH · 2025-10-28

**Soundness:** 3
**Presentation:** 3
**Contribution:** 3
**Rating:** 6
**Confidence:** 3

**Summary:**

This paper proposed a FL paradigm named FedDRM, that treats client heterogeneity as an asset rather than a limitation.
By leveraging an empirical likelihood-based framework, FedDRM simultaneously trains personalized local models and enables the central server to intelligently route new queries to the most suitable client.
However, the experiments are limited to CIFAR benchmarks, which are small scale and fewer than existing methods. They may be insufficent to demonstrate the superity of the proposed method.

**Strengths:**

1. The idea of leveraging heterogeneity for query routing instead of merely mitigating it, seems interesting and offer a novel perspective in FL.

2. The empirical likelihood framework provides a principled foundation for joint model training and client matching, with a tractable dual loss formulation.

**Weaknesses:**

1. The only dataset is CIFAR (including cifar 10 and cifar 100), which is significient different from the data in domains such as healthcare and finance. This limits the support for claims about the method’s potential in such areas.

2. The client-classification head may face challenges when scaling to very large client pools (let's say, >100 clients), as gradient drift could become worsen.

**Questions:**

1. Could the client-classification mechanism be distilled into a lighter-weight model to reduce inference overhead?

2. How could FedDRM adapt to dynamic client populations? For example, the data distributions evolve over time.

---

> ### Author Response · Authors · 2025-11-20
>
> Thank you for your thoughtful feedback. We appreciate the reviewer’s positive assessment of our core idea—leveraging heterogeneity for intelligent query routing—and the recognition that our empirical-likelihood formulation offers a principled and tractable framework. Below, we address the raised concerns and will incorporate the reviewer’s suggestions into the final version.
>
> - **Additional experiments in healthcare**. We thank the reviewer for this insightful comment. We agree that validating methods solely on synthetic heterogeneity can be limiting, and we appreciate the opportunity to clarify the motivation and strengthen the empirical evidence. To directly address your concern, we have conducted a new experiment on the real-world medical dataset called RETINA, which contains retinal fundus image datasets from different institutions, hence naturally fits into our FL framework. The results (see below for your reference) show that our method consistently outperforms baselines. This demonstrates that our approach is effective not just in synthetic settings but also under real-world heterogeneous conditions.
> | **Method** | **Average Accuracy** | **System Accuracy** |
> | ---------- | -------------------- | ------------------- |
> | Ditto      | 90.55±0.12           | 85.23±0.10          |
> | FedRep     | 90.75±0.10           | 84.72±0.10          |
> | FedBABU    | 89.85±0.12           | 89.80±0.12          |
> | FedPAC     | 90.70±0.00           | 84.67±0.00          |
> | FedALA     | 89.47±0.14           | 89.52±0.16          |
> | FedAS      | 91.38±0.12           | 83.54±0.13          |
> | ConFREE    | 90.88±0.12           | 84.37±0.10          |
> | FedAvgFT   | 89.92±0.18           | 89.65±0.25          |
> | FedProxFT  | 89.95±0.36           | 89.62±0.20          |
> | FedSAMFT   | 88.44±0.00           | 88.07±0.13          |
> | `FedDRM`   | **92.21±0.34**       | **91.21±0.34**      |
> - **Clarification on inference cost.** We thank the reviewer for this insightful comment. We would like to clarify that our motivating application involves medical centers, where the number of participating clients is modest. As a result, the inference cost of FedDRM’s client-classification module is small. For example, in the main experiment, the classification head accounts for only 1.17% (133,896/11,437,138) of the total parameters, making its contribution to the overall computation negligible. In typical healthcare settings, where the client population is not large, our current approach remains practical and reliable.
>
>     To ensure scalability in more demanding scenarios, our framework can be readily extended along two dimensions. **First, in terms of model complexity**, if one were to deploy a substantially larger or deeper classification head, standard techniques such as knowledge distillation could be employed to obtain a lighter-weight surrogate model for routing. **Second, regarding system scale**, when the number of clients becomes very large, even a distilled model may still face challenges due to the need for precise client-level classification. In such cases, our method can be integrated with clustering techniques. By first grouping clients into similar clusters and then replacing the fine-grained 'client classification' with a coarse-grained 'cluster classification,' we can effectively manage computational complexity. Exploring these directions is an interesting avenue for future work.

---

> ### Author Response · Authors · 2025-11-20
>
> - **Scalability to more clients.** We thank the reviewer for this insightful point. While scaling the client-classification head to very large client pools (e.g., >100 clients) may exacerbate gradient drift, techniques like those proposed in [1] could be incorporated to address this effectively. We can also apply clustering based idea as mentioned above to further resolve this issue. Importantly, our work introduces the novel perspective of leveraging heterogeneity for server-guided routing in FL, and we view scalability to larger client pools as an exciting and promising avenue for future research.
> - **Adapting FedDRM to dynamic client populations.** We thank the reviewer for this insightful question. In many healthcare applications, where clients correspond to hospitals, patient cohorts tend to remain relatively stable over time, making the current setting reasonable. To address evolving distributions, one promising direction would be to model each client with a mixture model capturing subpopulations that change over time. In this scenario, our exponential tilting approach could be extended to operate at the subpopulation level, allowing the server to dynamically adjust routing as client distributions drift. Other complementary approaches could include online updating of tilt parameters or temporal weighting of historical data to account for gradual shifts. We view this as an exciting avenue for future research to make FedDRM robust to evolving client populations.
>
> References:
>
> 1. Federated Learning with Only Positive Labels. ICML 2020

---

> > ### Comment · Reviewer_C2JH · 2025-11-25
> >
> > Thanks for the rebuttal. I would like to maintain my initial score and wait for feedback from the other reviewers before any further discussion.

---

> > > ### Author Response · Authors · 2025-11-26
> > >
> > > Thank you for the update. We appreciate your consideration and will be happy to clarify any additional points if needed.

---

### Official Review · Reviewer_Yf1W · 2025-10-31

**Soundness:** 2
**Presentation:** 2
**Contribution:** 2
**Rating:** 4
**Confidence:** 3

**Summary:**

This paper studies the problem of heterogeneity among clients in federated learning. The paper classifies the heterogeneity into two categories: i) Covariate shift ii) label shift. Then the paper investigates the problems of data heterogeneity in federated learning which are combination of these types heterogeneity.To address this problem, the paper defines new local loss functions for clients. Then using some adding heads to a base shared or global model, the federated learning is performed. Experiments on image classification is conducted to show the effectiveness of the proposed algorithms.

**Strengths:**

- The paper proposes more responsibilities for the server in federated learning such that the server guides the clients during training. This idea is interesting and can leads to improvements.
- The convergence rate of the proposed algorithm is obtained for strongly convex loss functions.
- The proposed algorithm is compared against variety of federated learning baselines.

**Weaknesses:**

- The most important weakness of the paper is presentation. The importance and novelty of the paper’s contributions are not clearly conveyed. I recommend that the authors explicitly and concisely list the main contributions of the work, such as the specific problems that the proposed algorithm aims to solve. At present, there is no clear evidence or justification demonstrating that the proposed algorithm effectively addresses the key issues claimed in the paper.
- The paper focuses solely on the multi-class classification problem, which is a relatively minor limitation.
- The discussion following Theorem 2.5 does not discuss if the proposed algorithm improves the convergence rate compared to existing methods.
- The data heterogeneity modeled in the experiments is synthetic. While this is common in federated learning research, the introduction suggests that the paper targets particular types of heterogeneity. Because of this, the practical applicability of the studied problem is not clear.

**Questions:**

- While reading the paper, I found it unclear why Theorem 2.1 is important. I suggest adding more explanations to clarify its significance, or alternatively, moving it to the appendix. Could you elaborate on this point?
- Could you also elaborate on the relevance of data modeling in the experiments to the problems discussed in the introduction?

---

> ### Author Response · Authors · 2025-11-20
>
> Thank you for your constructive feedback. We appreciate the reviewer’s recognition that assigning additional responsibilities to the server is an interesting and promising direction, as well as the acknowledgement of our convergence analysis and comparisons to a broad set of FL baselines. Below, we address all concerns in detail and will incorporate the reviewer’s suggestions into the final version.
>
> - **Clarification on the contribution.** We thank the reviewer for the careful reading and thoughtful feedback. To better highlight our contributions, we have revised the final two paragraphs of the introduction to make the contributions more explicit, strengthen the connection between the identified challenges and our algorithmic design choices, and add clearer signposting in the experimental section showing how the results support each contribution. We hope these changes help address the reviewer’s concerns and more effectively communicate the strengths of our work.
> - **Clarification on Theorem 2.1.** We thank the reviewer for raising this point. Theorem 2.1 plays an important role in our framework: it characterizes the exact form of each client’s marginal distribution relative to the reference distribution. This characterization is what enables us to construct the joint distribution and derive the log-likelihood expression used in Line 188, which in turn underpins the formulation of our proposed objective. In other words, the theorem provides the structural foundation that allows our method to handle distributional heterogeneity in a principled way. To improve clarity, we have added explanatory text immediately following Theorem 2.1, as well as additional commentary around Line 210, explicitly highlighting how the theorem’s result is used to derive the likelihood and how it supports the subsequent algorithmic design. We hope this makes the significance of the theorem more evident and the logical flow easier to follow.
> - **Convergence results.** We thank the reviewer for this question. Theorem 2.5 does not claim that our algorithm achieves a faster convergence rate than existing methods. Instead, it characterizes the trade-off between convergence and accuracy as a function of the regularization parameter $\lambda$. This trade-off is empirically illustrated in Figure 4, where we explicitly highlight its connection to Theorem 2.5. We have revised the discussion to make the connection clearer in the manuscript.
> - **Clarification on modelling approach in the experiments.** We thank the reviewer for this comment. As discussed in the introduction, client heterogeneity can arise from differences in feature distributions (covariate shift), label distributions (label shift), or both. In our CIFAR experiments, we emulate covariate shift by applying nonlinear image transformations. These transformations preserve the semantic content and ground-truth labels—i.e., they do not alter the object depicted—but they do modify low-level appearance attributes such as color balance and illumination. This induces changes in the pixel distribution and thus corresponds to covariate shift in our setting. To model label shift, we use either a Dirichlet distribution or a 5-SPC allocation scheme to generate heterogeneous label proportions across clients. Because these sampling schemes change only the frequency with which each label appears on each client while leaving the underlying data generation process $p(x \mid y)$ unchanged, the resulting heterogeneity is precisely label shift.
> - **Clarification on problem setting.** We appreciate the reviewer’s comment. It is correct that our method focuses on multi-class classification; however, classification is one of the core problems in supervised learning and remains the standard and most widely used evaluation setting in federated learning, particularly for studying data heterogeneity. Our problem formulation is motivated by healthcare applications, where classification is a primary task, and our contribution—advocating for an active FL system—addresses a practically important and under explored aspect of this setting. Extending our approach to other supervised tasks is an interesting direction for future work, and we will consider this in subsequent research.

---

> ### Author Response · Authors · 2025-11-20
>
> - **Experiment on real dataset.** We thank the reviewer for this insightful comment regarding using benchmark dataset for evaluation. We agree that validating methods solely on synthetic heterogeneity can be limiting, and we appreciate the opportunity to clarify the motivation and strengthen the empirical evidence. To directly address your concern, we have conducted a new experiment on the real-world medical dataset called RETINA, which contains retinal fundus image datasets from different institutions, hence naturally fits into our FL framework. The results (see below for your reference) show that our method consistently outperforms baselines. This demonstrates that our approach is effective not just in synthetic settings but also under real-world heterogeneous conditions.
> | **Method** | **Average Accuracy** | **System Accuracy** |
> | ---------- | -------------------- | ------------------- |
> | Ditto      | 90.55±0.12           | 85.23±0.10          |
> | FedRep     | 90.75±0.10           | 84.72±0.10          |
> | FedBABU    | 89.85±0.12           | 89.80±0.12          |
> | FedPAC     | 90.70±0.00           | 84.67±0.00          |
> | FedALA     | 89.47±0.14           | 89.52±0.16          |
> | FedAS      | 91.38±0.12           | 83.54±0.13          |
> | ConFREE    | 90.88±0.12           | 84.37±0.10          |
> | FedAvgFT   | 89.92±0.18           | 89.65±0.25          |
> | FedProxFT  | 89.95±0.36           | 89.62±0.20          |
> | FedSAMFT   | 88.44±0.00           | 88.07±0.13          |
> | `FedDRM`   | **92.21±0.34**       | **91.21±0.34**      |

---

> ### Author Response · Authors · 2025-11-26
>
> Dear Reviewer Yf1W,
>
> We hope you are doing well. With the discussion phase concluding in **one week**, we would like to respectfully check if our revised paper and the additional experiments have sufficiently resolved your concerns. Please let us know if there are any remaining points or further feedback we should address. We are fully committed to improving our work based on your insights.
>
> Thank you again for your time and effort.

---

### Official Review · Reviewer_avg5 · 2025-10-31

**Soundness:** 3
**Presentation:** 3
**Contribution:** 3
**Rating:** 4
**Confidence:** 3

**Summary:**

This paper proposes FedDRM, a federated learning (FL) framework that extends beyond traditional aggregation to enable the central server to actively route new queries (e.g., patient data) to the most suitable client in heterogeneous environments. Drawing from a healthcare motivation—directing patients to specialized hospitals—the authors model client data distributions using a semiparametric density ratio model (DRM) anchored to a nonparametric empirical likelihood (EL) baseline at the server. Under assumptions like covariate shift, they derive exponential tilts linking client distributions (Eqs. 2-3) and extend this to marginals (Theorem 2.1). The profiled EL loss decomposes into cross-entropy terms for label prediction and client identification, jointly optimizing local accuracy and routing precision. Experiments on CIFAR-10/100 with Dirichlet heterogeneity (α=0.3) compare against baselines like FedAvg, reporting gains in "system acc" (routed performance) and guidance metrics, though results appear inconsistent across tables.

**Strengths:**

1. The core idea of reframing heterogeneity as an asset—via server-guided routing— is a genuine advance over passive FL methods. By turning the server into an "intelligent router" (Fig. 1), it addresses a practical gap: e.g., in finance, routing fraud queries to distribution-matched banks, or in healthcare, assigning patients to expert clients. This not only boosts efficiency but also highlights FL's untapped potential for adaptive, privacy-preserving systems.

2. The DRM-EL framework elegantly unifies heterogeneous training with query matching. The use of multiplicative tilts (Eq. 3) to capture shifts without raw data sharing is nonparametric yet efficient, and the loss decomposition (label + client CE) provides a clean, interpretable objective. Theorem 2.1's extension to marginals under covariate shift adds rigor, making the method theoretically grounded while extensible to label/full shifts.

3. Despite limitations, the reported "system acc" improvements (e.g., leveraging routing for end-to-end gains) demonstrate the framework's promise over aggregation-only baselines. This dual focus—accuracy + precision—positions FedDRM as a step toward resource-efficient FL, with broad applicability in multi-client domains.

**Weaknesses:**

1. The introduction builds excitement around the routing idea but omits any preview of the DRM or EL machinery, leaving readers disoriented when section 2 dives straight into FedDRM. What is DRM? A brief 1-2 sentence teaser in the intro would bridge this.

2. While results show promise, the scope is too narrow to substantiate claims under severe heterogeneity. You only report Dirichlet (dir) α=0.3 (low heterogeneity; near-IID), but FL's real pain is high heterogeneity. How does FedDRM scale here? Add analysis across α values or extend to more datasets to show robustness.

3. What is the number of clients in the two main tables? I found that the results seem best fit with m=8; however, the results are not well-aligned. The system acc for FedDRM is 59.59% under dir=0.3, which is significantly lower than the result in the first main table.

4. FedDRM requires uploading additional density ratio parameters (e.g., tilts γ_i, ξ_i in Eq. (3), lines 144-147) alongside model updates, potentially undermining FL's privacy guarantees (e.g., vs. FedAvg's aggregated-only sharing, lines 32-35). These tilts encode distributional shifts (e.g., π_i margins), risking inference attacks on sensitive traits like demographics (Fig. 1, lines 45-60).

**Questions:**

1. The authors only report the results when Dirichlet (dir) α=0.3 (low heterogeneity; near-IID), but FL's real pain is high heterogeneity. How does FedDRM scale here? Could the authors add analysis across more α values or extend to more datasets to show robustness?
2. Could you please explain the experimental details? For example, what is the number of clients in the two main tables? I found that the results seem best fit with m=8; however, the results are not well-aligned.
3. Could the author please provide privacy guarantees for the additional uploaded parameters?

---

> ### Author Response · Authors · 2025-11-20
>
> We sincerely thank the reviewer for your constructive feedback. We’re encouraged that you found our formulation of heterogeneity-aware routing to be a meaningful and novel contribution, and that the DRM–EL framework and its empirical gains were viewed positively. Below, we address your comments in detail and have incorporate all suggested improvements into the revised version.
>
> - **Brief introduction about DRM and EL.** We thank the reviewer for the constructive feedback on improving the clarity of our paper. In the revised version, we have added a brief introduction to DRM and EL both in the introduction and the second section where we introduced DRM rigorously for the first time in the paper. The changes are highlighted in blue for easy reference.
> - **Clarification on the experimental results in Table 1/2 and Table 3.** We thank the reviewer for this question. To clarify, the main result Table 1/2 and the sensitivity analysis Table 3 present results under two distinct covariate shift settings, which are detailed in Figures 8 and 10 in the appendix. Specifically, the covariate shift in Table 3 is less severe than in Table 1/2, leading to lower system accuracy but higher average accuracy—when shifts are milder, distinguishing between clients becomes more challenging. Nonetheless, our method consistently outperforms all baselines under the Table 3 setting. For completeness, we further provide the comparison results for all methods with $m=8$ under the Table 3 setting:
>
>     | **Method** | **System Accuracy** | **Average Accuracy** |
>     | ---------- | ------------------- | -------------------- |
>     | Ditto      | 45.67±0.21          | 76.03±0.16           |
>     | FedRep     | 28.59±0.19          | 76.26±0.13           |
>     | FedBABU    | 54.13±0.18          | 77.87±0.14           |
>     | FedPAC     | 28.57±0.20          | 76.26±0.11           |
>     | FedALA     | 57.30±0.18          | 62.65±3.98           |
>     | FedAS      | 32.58±0.19          | 78.86±0.12           |
>     | ConFREE    | 29.48±0.21          | 76.62±0.12           |
>     | FedAvgFT   | 53.17±0.22          | 78.92±0.15           |
>     | FedProxFT  | 53.72±0.21          | 79.06±0.14           |
>     | FedSAMFT   | 51.89±0.20          | 75.81±0.10           |
>     | `FedDRM`   | **59.59±0.20**      | **80.47±0.12**       |
> - **Additional experiments on membership inference attack in terms of privacy guarantee.** We thank the reviewer for raising this important question about privacy risks. In response, we conducted additional experiments to evaluate membership inference attacks (MIA) under FedDRM. We consider two standard threat models: (i) a relatively mild setting where the attacker only has access to logits, and (ii) a stronger adversary with access to full gradient information. We compare two schemes: (a) our proposed FedDRM, and (b) FedDRM without the machine-classification head (removing the main additional model information related to tilting term). Across both threat models, we find that FedDRM exhibits only a **marginally higher** MIA risk than FedDRM (partial) when no additional protection is applied. Importantly, the increase is small and does not indicate a practically meaningful escalation in information leakage. Moreover, because FedDRM is fully parameterized in the same way as standard FL methods, conventional privacy-preserving mechanisms—such as differential privacy, gradient clipping, or secure aggregation—can be directly incorporated to protect the uploaded parameters, including the density-ratio tilts. Furthermore, we can also follow the technique in [1] to only share the client corresponding class embedding weight for machine classification head to further avoid this risk.
>
>     |   Method     | Metric | Logit | Gradient |
>     |--------|-----|-------|----------|
>     | FedDRM| ACC | 0.6643| 0.5552 |
>     |                    | AUC | 0.5097| 0.5809 |
>     | FedDRM(partial)| ACC | 0.6622| 0.5519 |
>     |                    | AUC | 0.5064| 0.5785 |

---

> ### Author Response · Authors · 2025-11-20
>
> - **Severeness of heterogeneity.** We thank the reviewer for the question. We have added a **sensitivity analysis** comparing our method with the baselines across a range of Dirichlet parameters $\alpha$. Consistent with prior work [2], we observe that smaller $\alpha$ values—corresponding to higher data heterogeneity—lead to higher *average* local accuracy for all methods. This occurs because each client’s training and testing data are drawn from the same distribution. As $\alpha$ decreases, the local label distributions become increasingly skewed, with some classes receiving negligible probability mass. This effectively reduces the number of classes present on each client, thereby simplifying the local classification problem relative to the balanced case. For system-level accuracy, we find that this trend persists for our method, as the only additional component is the client-routing step, which does not alter the underlying behavior of local classification. In contrast, methods such as FedRep exhibit increasing system accuracy as $\alpha$ grows (i.e., as the label distributions become more homogeneous). When $\alpha$ is small, the local models become highly personalized and fail to reach a consistent consensus across clients, causing majority voting to misroute queries and thus lowering system accuracy. As $\alpha$ increases, this inconsistency diminishes, and the aggregated routing accuracy improves. These results further confirm that our method remains robust across varying degrees of label shift.
>
>     Moreover, we would like to clarify a potential misunderstanding regarding the Dirichlet parameter setting. In the standard FL literature [3], a smaller $\alpha$ indicates a more severe label shift. Therefore, our setting of $\alpha=0.3$ represents a highly heterogeneous scenario, rather than a near-IID one. Specifically, under this setting, each client is dominated by only 2-4 classes. We have provided visualizations of the label distributions across clients under different $\alpha$ values in Figure 9 and Figure 14.
>
>
>     | **Average Acc** | **α=0.3**      | **α=1.3**      | **α=2.3**      |
>     | --------------- | -------------- | -------------- | -------------- |
>     | Ditto           | 76.34±0.11     | 55.56±0.18     | 52.76±0.18     |
>     | FedRep          | 76.49±0.15     | 55.09±0.17     | 52.95±0.16     |
>     | FedBABU         | 78.22±0.14     | 61.61±0.16     | 59.83±0.20     |
>     | FedPAC          | 76.53±0.13     | 55.08±0.15     | 52.95±0.16     |
>     | FedALA          | 64.35±2.40     | 49.14±0.62     | 47.29±0.50     |
>     | FedAS           | 78.69±0.17     | 59.83±0.19     | 58.17±0.18     |
>     | ConFREE         | 76.73±0.16     | 55.56±0.18     | 53.58±0.15     |
>     | FedAvgFT        | 79.08±0.11     | 62.87±0.16     | 61.94±0.17     |
>     | FedProxFT       | 79.07±0.12     | 62.84±0.15     | 61.86±0.19     |
>     | FedSAMFT        | 75.53±0.11     | 55.46±0.19     | 53.54±0.15     |
>     | `FedDRM`        | **80.25±0.14** | **64.04±0.16** | **62.30±0.15** |
>
>
>
>     | **System Acc** | **α=0.3**      | **α=1.3**      | **α=2.3**      |
>     | -------------- | -------------- | -------------- | -------------- |
>     | Ditto          | 47.64±0.25     | 43.78±0.23     | 44.04±0.20     |
>     | FedRep         | 24.96±0.19     | 35.45±0.22     | 37.87±0.22     |
>     | FedBABU        | 57.43±0.17     | 54.60±0.19     | 54.25±0.20     |
>     | FedPAC         | 25.14±0.21     | 35.53±0.21     | 37.84±0.20     |
>     | FedALA         | 61.33±0.17     | 48.47±0.18     | 46.68±0.20     |
>     | FedAS          | 28.76±0.19     | 41.43±0.20     | 46.08±0.22     |
>     | ConFREE        | 25.66±0.22     | 36.12±0.21     | 38.78±0.23     |
>     | FedAvgFT       | 54.90±0.22     | 54.80±0.18     | 55.61±0.19     |
>     | FedProxFT      | 55.01±0.20     | 54.84±0.15     | 55.62±0.18     |
>     | FedSAMFT       | 55.83±0.21     | 49.15±0.17     | 47.71±0.16     |
>     | `FedDRM`       | **63.85±0.18** | **56.83±0.18** | **56.00±0.23** |
>
>
> References:
>
> 1. Federated Learning with Only Positive Labels. ICML 2020
> 2. Personalized Federated Learning with Feature Alignment and Classifier Collaboration. ICLR 2023
> 3. Bayesian Nonparametric Federated Learning of Neural Networks. ICML 2019

---

> ### Author Response · Authors · 2025-11-26
>
> Dear Reviewer avg5,
>
> We hope this message finds you well. As the discussion phase has entered its **final week**, we would like to confirm if our revised paper and the additional experiments have addressed all your concerns satisfactorily. If there are any additional points or feedback you would like us to consider, please let us know. Your insights are invaluable to us, and we are eager to address any remaining issues to improve our work.
>
> Thank you for your time and effort in reviewing our paper.

---

### Meta-Review · Area_Chair_faE5 · 2026-01-04

**Summary:**

This paper proposes FedDRM, a federated learning framework that leverages client heterogeneity as a resource. Unlike standard FL methods that passively aggregate updates, the server in FedDRM actively routes new queries to the most appropriate client. This is enabled by a Density Ratio Model (DRM) combined with an Empirical Likelihood (EL) framework, which jointly optimizes local model accuracy and routing precision. The authors provide convergence guarantees for strongly convex losses and evaluate the method on CIFAR-10/100 as well as a real-world medical dataset (RETINA). Empirical results demonstrate clear improvements in both system-level accuracy and routing effectiveness compared to existing FL baselines.

**Reviewer Concerns:**

- Theoretical clarity and significance
Reviewers questioned the role of Theorems 2.1 and 2.5 and how the proposed framework improves or generalizes standard FL guarantees. Authors clarified that the theorems underpin the method’s principled handling of heterogeneity.

- Experimental scope and heterogeneity
Concerns were raised regarding the degree of heterogeneity and applicability to real-world scenarios. The authors responded by adding sensitivity analyses across multiple Dirichlet parameters and including experiments on the RETINA medical dataset, demonstrating robustness under varied heterogeneity.

- Scalability and practical deployment
Scaling the client-classification head to large or dynamic client populations was noted as a potential challenge. Authors discussed practical solutions such as clustering and knowledge distillation, indicating that the approach remains feasible in typical healthcare settings.

**Reviewer Scores:**

Reviewer avg5. The authors provide extensive additional experiments and clarify the effect of varying Dirichlet parameters, as well as membership inference risks, which address the reviewer’s main concerns regarding heterogeneity and privacy. The reviewer may be likely to keep the score.

Reviewer Yf1W. The authors clarified the contributions, explained the significance of Theorem 2.1, and added experiments on the RETINA dataset to demonstrate practical applicability. These revisions improve the clarity and completeness of the paper, and the reviewer may likely to maintain the original score.

Reviewer C2JH.
The authors largely address this reviewer’s concerns, and the reviewer is likely to maintain the score.

---

### Decision · Program_Chairs · 2026-01-26

Accept (Poster)